# Stepwise Guided Policy Optimization: Coloring Your Incorrect Reasoning in GRPO

**Peter Chen**                                                  *lc3826@columbia.edu*
*Department of Mathematics*
*Columbia University*

**Xiaopeng Li**                                           *xiaopengli1@cuhk.edu.cn*
*School of Artificial Intelligence*
*The Chinese University of Hong Kong, Shenzhen*

**Ziniu Li**                                                *liziniu1997@gmail.com*
*School of Data Science*
*The Chinese University of Hong Kong, Shenzhen*

**Xi Chen**                                                   *xc13@stern.nyu.edu*
*Stern School of Business*
*New York University*

**Tianyi Lin**                                               *tl3335@columbia.edu*
*Department of Industrial Engineering and Operation Research*
*Columbia University*

**Reviewed on OpenReview:** *https://openreview.net/forum?id=ALnVAqtshR*

## Abstract

Reinforcement learning (RL) has proven effective in strengthening the reasoning capabilities of large language models (LLMs). A widely adopted method, Group Relative Policy Optimization (GRPO) (Shao et al., 2024), has shown strong empirical results in training recent reasoning models (Guo et al., 2025a), but it fails to update the policy when all responses within a group are incorrect (i.e., all-negative-sample groups). This limitation highlights a gap between artificial and human intelligence: unlike humans, who can learn from mistakes, GRPO discards these failure signals. We introduce a simple framework to mitigate the all-negative-sample issue by incorporating response diversity within groups using a *step-wise* judge model, which can be trained directly or adapted from existing LLMs. In a simplified setting, we prove that this diversification accelerates GRPO's learning dynamics. We then empirically validate Stepwise Guided Policy Optimization (SGPO) across model sizes (7B, 14B, 32B) in both offline and online training on nine reasoning benchmarks (including base and distilled variants). Overall, SGPO improves average performance and is effective in early and mid-training when all-negative groups are prevalent, while improvements are not uniform across every benchmark and depend on the structure and informativeness of negative samples. Finally, SGPO does not require the judge model to generate correct solutions, distinguishing it from knowledge distillation methods.

## 1 Introduction

The rise of OpenAI-o1 (Jaech et al., 2024), DeepSeek-R1 (Guo et al., 2025a), and Kimi-1.5 (Team et al., 2025) has highlighted the emergence of *large AI reasoning models.* Unlike instruction-tuned models (Brown et al., 2020; Chowdhery et al., 2023; Touvron et al., 2023; Achiam et al., 2023), which produce quick responses by

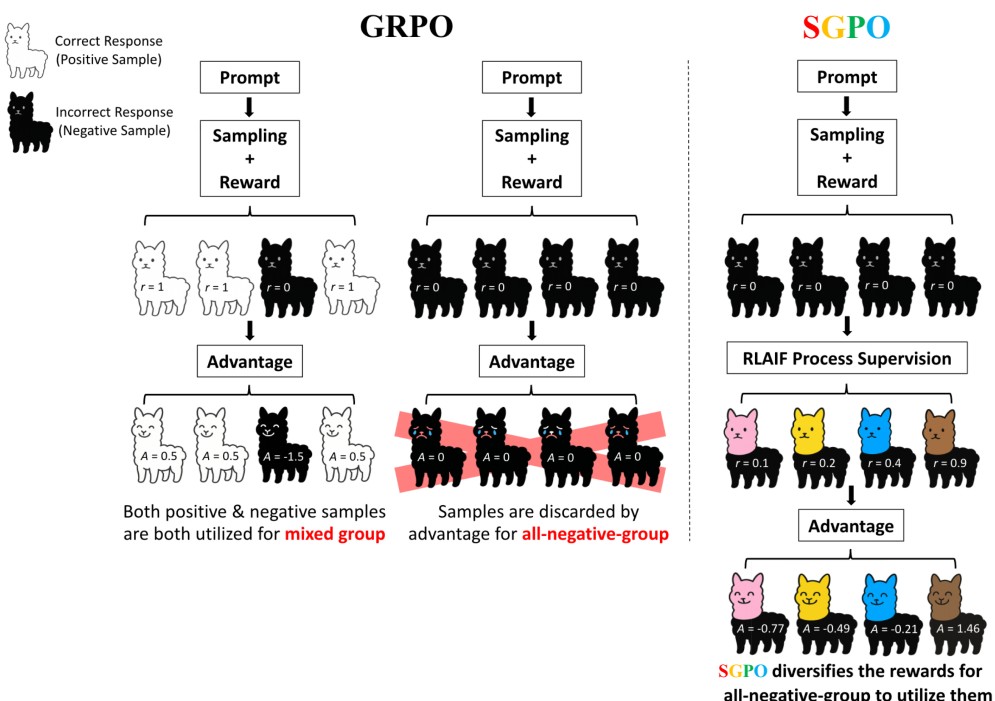

Figure 1: Main pipeline for Stepwise Guided Policy Optimization. On each llama, $r$ indicates the reward of the sampled response and $A$ indicates response's advantage through group relative computation.

statistically inferring the next token, these new reasoning models deliberately decompose complex prompts (e.g., mathematical problems) into intermediate steps and work through chain-of-thought reasoning (Wei et al., 2022; Yao et al., 2023; Besta et al., 2024; Xiang et al., 2025). This slower yet more rigorous process yields greater accuracy and makes them more human-like, enabling success on more complex and challenging tasks (Yang et al., 2018; Shi et al., 2024; Jain et al., 2025). As generative AI applications move beyond single-turn chat and question-answering, these reasoning models are poised to become more powerful and widely adopted, positioning them as a foundational component of modern AI systems.

At the heart of this revolution lies post-training with outcome-based and verifiable rewards (Cobbe et al., 2021; Uesato et al., 2022; Zelikman et al., 2022; Singh et al., 2023; Hosseini et al., 2024; Lightman et al., 2024; Wang et al., 2024; Setlur et al., 2025; Zhang et al., 2025b), together with reinforcement learning (RL) methods (Schulman et al., 2015; 2017; Li et al., 2024b; Ahmadian et al., 2024; Shao et al., 2024; Xiong et al., 2025a), appreciated for their simplicity, intuitiveness, and practicality. A leading approach is proximal policy optimization (PPO) (Schulman et al., 2017), which relies on a critic (or value) model to estimate advantages. While essential in general RL tasks, this critic is often unnecessary in large language models (LLMs) due to their deterministic transition dynamics (Li et al., 2024b). This observation has inspired alternatives such as group relative policy optimization (GRPO) (Shao et al., 2024) and its extensions (Yu et al., 2025b; Liu et al., 2025b; Chu et al., 2025; Zhang et al., 2025a), which estimate advantages directly in a group-relative fashion (normalizing rewards across multiple samples for the same prompt).

A major limitation of these methods arises when all sampled responses in a group are incorrect (i.e., *all-negative-sample* groups), which eliminates the learning signal and halts policy updates. In GRPO, given a prompt $\mathbf{x}$, responses $\{\mathbf{y}_i\}_{i=1}^G$ are drawn from the old policy $\pi_{\text{old}}$ and assigned rewards $\{r_i\}_{i=1}^G$, where $r_i = 1$ if $\mathbf{y}_i$ is correct and 0 otherwise. Advantages are obtained by normalizing $r_i$ within the group. If $r_i = 0$ for all $i$, the advantage vanishes, yielding no update. Such groups are frequent in early and mid-stages of training, when reasoning ability is weak[1]. This shortcoming highlights a gap between artificial and human intelligence: humans effectively learn from mistakes, which act as essential signals during cognitive development (Chialvo

---

[1]To reduce computational cost, training often uses small group sizes and short rollouts, further increasing the likelihood of all-negative-sample groups.

& Bak, 1999). In mathematical reasoning, all-negative-sample groups prompt a child to revise rules and strengthen reasoning ability.

Recent studies suggest that negative samples in RL-based large reasoning model training carry more nuanced value than previously assumed (Xiong et al., 2025a). Instead of treating negative samples uniformly, they advocate for principled mechanisms to distinguish negative samples. One prominent direction is process reward models (PRMs) (Lightman et al., 2024; Wang et al., 2024; Luo et al., 2024; Setlur et al., 2025; Zhang et al., 2025b), which estimate either the probability of final success or its change after each reasoning step. However, their reliance on speculative value functions makes them prone to reward hacking (Skalse et al., 2022).

A common observation is that many reasoning tasks possess a structure where step-level correctness can be explicitly defined. This motivates the use of a step-wise judge model that evaluates trajectories by labeling each step as correct (1) or incorrect (0). Such a model can be trained directly (Xiong et al., 2025b) or adapted from existing LLMs (Zha et al., 2025; He et al., 2025)[2]. By grounding rewards in step-level correctness rather than speculative value estimates, our method mitigates reward hacking and yields clearer signals. Intuitively, this allows negative samples to be differentiated through their trajectories: while early-stage reasoning trajectories are of low-quality, these remain informative – much like partial credit in education, where intermediate steps still guide learning.

Our approach enables a holistic evaluation of multi-step reasoning by transforming negative samples from binary outcome rewards into graded, step-level rewards. Consider a negative sample with five reasoning steps $(a_1, a_2, a_3, a_4, a_5)$. If the first error occurs at $a_3$, then $a_1$ and $a_2$ are correct, yielding a correctness proportion of $\frac{2}{5}$. To improve reliability, we adopt a Grok4-Heavy -inspired strategy where multiple independent judgments are obtained from the judge model, and the error position is determined by the majority vote. We further introduce two scaling parameters $\beta$ and $\gamma$ to downweight noisy or unreliable signals (see Eq. (2)). Unlike PRMs, our approach avoids memory overhead and does not require costly step-level human annotations, thereby accelerating training. In this work, we focus on outcome-based post-training with group-relative updates (GRPO-style) for structured reasoning tasks; extending our approach to arbitrary reward settings is beyond the scope of this paper and left to future work.

**Contribution.** We propose and analyze a simple and efficient framework that introduces response diversity within all-negative-sample groups. It is both theoretically grounded in the simplified setting and empirically effective on various models, distinguishing our approach from existing heuristics. Our contributions can be summarized as follows:

1. We propose a *Stepwise Guided Policy Optimization* (SGPO) framework that leverages a step-wise judge model that identifies the first incorrect step that causes the trajectory to deviate from correctness. This makes evaluation computationally tractable and reliable. *It is important to emphasize that our contribution lies not in designing effective judge models, but in introducing a framework that leverages step-wise judges to effectively distinguish negative samples.* We also prove that SGPO outperforms GRPO in a simplified setting.

2. We conduct experiments demonstrating the effectiveness of our approach in improving LLM reasoning. Evaluations are undertaken across various model sizes (7B, 14B, 32B) in both offline and online settings with nine benchmarks, including base and distilled variants, using GRPO as the primary baseline given our focus on outcome-based group-relative RLVR. Our results reveal two key benefits: (i) SGPO delivers improvements beyond the reach of GRPO, especially in the early and mid-stages of training where all-negative-sample groups are common; (ii) SGPO does not rely on more powerful judge models generating correct answers, allowing it to be distinguished from knowledge distillation.

The additional overhead from all-negative-sample groups remains modest, since the correctness can be efficiently verified against reference solutions, enabling rapid assessment of reasoning steps. As the computational and financial costs of closed-source judge models (`o4-mini`, `Claude3.7`) rise, SGPO accelerates learning dynamics, making the trade-off worthwhile. SGPO also outperforms GRPO with less powerful and

---

[2]We do not have access to their judge models as it's not publicly released, so we adapt our own from existing LLMs.

more affordable open-source judge models (`DeepSeek-V3-0324`, `Qwen3-235B-A22B`, `QwQ-32B`), confirming that SGPO remains effective even without cutting-edge LLMs and underscoring its practicality in lower-resource settings.

## 2 Preliminaries and Technical Background

**LLM finetuning.** LLM finetuning typically consists of *pre-training* and *post-training*. Pre-training equips the model with broad language understanding and generation capabilities, while post-training adapts the model to specific downstream objectives (e.g., improving mathematical problem solving via reasoning). In post-training, a model usually first undergoes *imitation learning* (e.g., supervised finetuning on human or expert trajectories, or direct distillation from stronger models), and then further improves by training on self-generated responses paired with feedback indicating whether the responses are good or bad. Two common feedback-driven paradigms are *reinforcement learning from human feedback* (RLHF), where a learned reward model (trained from human preferences) assigns rewards to model outputs, and *reinforcement learning from verifiable rewards* (RLVR), where rewards are computed by an automatic verifier (often used for math problems with checkable answers). We introduce GRPO, a representative RLVR method, below.

**Policy optimization.** Modern LLMs are built based on the Transformer architecture (Vaswani et al., 2017) and generate responses $\mathbf{y} = (a_1, \ldots, a_H)$ to user prompts $\mathbf{x}$, where each token $a_h \in \mathcal{V}^\star$, with $\mathcal{V}$ denoting the vocabulary and $\mathcal{V}^\star$ the set of all possible token sequences. We view the LLM as a policy $\pi_\theta(\mathbf{y}|\mathbf{x})$ parameterized by $\theta$, assigning probabilities to responses $\mathbf{y}$ given $\mathbf{x}$. The policy operates in an auto-regressive way as follows (Agarwal et al., 2020; Mei et al., 2021; Li et al., 2024b):

$$\pi_\theta(\mathbf{y}|\mathbf{x}) = \prod_{h=1}^{H} \pi_\theta(a_h \mid \mathbf{x}, a_1, \ldots, a_{h-1}).$$

For a prompt $\mathbf{x}$ with ground-truth response $\mathbf{y}_\mathbf{x}^\star$, performance is evaluated using a regular-expression match on the final answer: $r(\mathbf{x}, \mathbf{y}) = 1$ if $\mathbf{y}$ matches $\mathbf{y}_\mathbf{x}^\star$ and $r(\mathbf{x}, \mathbf{y}) = 0$ otherwise (Hendrycks et al., 2021). We consider the reasoning tasks defined over a dataset $\mathcal{D} = (\mathbf{x}, \mathbf{y}_\mathbf{x}^\star)$, where each $\mathbf{x}$ is a problem and $\mathbf{y}_\mathbf{x}^\star$ its ground-truth solution.

The policy gradient methods (Williams, 1992; Sutton & Barto, 1998) aim to maximize the objective $J(\theta) = \mathbb{E}_{\mathbf{x} \sim \rho, \mathbf{y} \sim \pi_\theta(\cdot|\mathbf{x})}[r(\mathbf{x}, \mathbf{y})]$ where $\rho$ is the prompt distribution and $\pi_\theta$ is an LLM policy. Parameters are updated via $\theta \leftarrow \theta + \eta \nabla_\theta J(\theta)$. In practice, trajectories are sampled from an old policy $\pi_{\theta_{\text{old}}}$, which is different from $\pi_\theta$, motivating the use of importance sampling as follows:

$$J(\theta) = \mathbb{E}_{\mathbf{x} \sim \rho, \mathbf{y} \sim \pi_{\theta_{\text{old}}}(\cdot|\mathbf{x})} \left[ \frac{\pi_\theta(\mathbf{y}|\mathbf{x})}{\pi_{\theta_{\text{old}}}(\mathbf{y}|\mathbf{x})} r(\mathbf{x}, \mathbf{y}) \right].$$

However, this estimator suffers from high variance when $\pi_\theta$ deviates from $\pi_{\theta_{\text{old}}}$. To stabilize training, clipped surrogate objectives are used as follows:

$$J(\theta) = \mathbb{E}_{\mathbf{x} \sim \rho, \mathbf{y} \sim \pi_{\theta_{\text{old}}}(\cdot|\mathbf{x})} \left[ \min \left\{ \frac{\pi_\theta(\mathbf{y}|\mathbf{x})}{\pi_{\theta_{\text{old}}}(\mathbf{y}|\mathbf{x})} r(\mathbf{x}, \mathbf{y}), \texttt{clip} \left( \frac{\pi_\theta(\mathbf{y}|\mathbf{x})}{\pi_{\theta_{\text{old}}}(\mathbf{y}|\mathbf{x})}, 1 - \epsilon, 1 + \epsilon \right) r(\mathbf{x}, \mathbf{y}) \right\} \right],$$

where $\texttt{clip}(x, 1 - \varepsilon, 1 + \varepsilon) := \max\{\min\{x, 1 + \varepsilon\}, 1 - \varepsilon\}$. The group relative policy optimization (GRPO) and its variants (Yu et al., 2025b; Liu et al., 2025b; Chu et al., 2025; Zhang et al., 2025a) adopt this framework but estimate gradients using groups of samples. For each prompt $\mathbf{x}$, GRPO samples responses $\mathbf{y}_1, \ldots, \mathbf{y}_G$ from $\pi_{\theta_{\text{old}}}$. We aim at maximizing the objective function in the form of

$$J(\theta) = \mathbb{E}_{\mathbf{x} \sim \rho, \{\mathbf{y}_i\}_{i=1}^{G} \sim \pi_{\theta_{\text{old}}}(\cdot|\mathbf{x})} \left[ \frac{1}{G} \sum_{i=1}^{G} \min \left\{ \frac{\pi_\theta(\mathbf{y}_i|\mathbf{x})}{\pi_{\theta_{\text{old}}}(\mathbf{y}_i|\mathbf{x})} A_i, \texttt{clip} \left( \frac{\pi_\theta(\mathbf{y}_i|\mathbf{x})}{\pi_{\theta_{\text{old}}}(\mathbf{y}_i|\mathbf{x})}, 1 - \epsilon, 1 + \epsilon \right) A_i \right\} \right],$$

where $\epsilon \in (0, 1)$ and the advantage $A_i$ is computed as

$$A_i = \frac{r(\mathbf{x}, \mathbf{y}_i) - \texttt{mean}(\{r(\mathbf{x}, \mathbf{y}_1), \ldots, r(\mathbf{x}, \mathbf{y}_G)\})}{\texttt{std}(\{r(\mathbf{x}, \mathbf{y}_1), \ldots, r(\mathbf{x}, \mathbf{y}_G)\})}, \tag{1}$$

where $r(\mathbf{x}, \mathbf{y}_i) = 1$ if $\mathbf{y}_i$ matches the ground-truth answer and 0 otherwise.

**Remark 2.1.** *When rewards are identical across all samples within a group, $A_i = 0$ and no update occurs. This is appropriate for all-positive groups but constitutes a critical limitation for all-negative groups, where GRPO fails to exploit mistakes as learning signals.*

## 3 Main Results

We propose the Stepwise Guided Policy Optimization (SGPO) framework, which employs the step-wise judge model to detect the first incorrect step that leads a trajectory away from correctness. In a simplified setting, we prove that SGPO consistently accelerates GRPO's learning dynamics.

### 3.1 A Step-wise Judge Model

We propose a principled reward mechanism for negative samples, wherein the step-wise judge model differentiates between structurally sound but partially incorrect reasoning and entirely erroneous responses. This design is motivated by the intuition that an incorrect final answer does not invalidate the entire reasoning process. For instance, a model may follow a logically coherent sequence of steps yet make a minor error – such as an arithmetic slip – that leads to an incorrect conclusion. Treating such cases the same as fundamentally flawed or incoherent reasoning does not make sense. This refinement remains effective under constraints such as reduced output length, where a model may be unable to complete the full solution but still demonstrates a valid reasoning trajectory.

Our step-wise judge model can be either trained directly or adapted from existing LLMs. It evaluates responses sequentially, identifying the first substantive error – such as a computational slip or a logical fallacy – that causes the trajectory to deviate from correctness. To formalize this, we define the *Reasoning Trajectory Score* (RTS) for an incorrect response $\mathbf{y}$, denoted as $\texttt{RTS}(\mathbf{y}) \in [0, 1]$. The judge model checks each step in order, pinpoints the first error, and treats all preceding steps as the valid reasoning segment. $\texttt{RTS}(\mathbf{y})$ is then computed as the ratio of the valid segment length to the total trajectory length. For example, if $\mathbf{y}$ consists of five steps $(a_1, a_2, a_3, a_4, a_5)$ and the first error occurs at $a_4$, then $\texttt{RTS}(\mathbf{y}) = \frac{3}{5}$, indicating that three steps of reasoning are correct before erroneous.

In our experiment, we adapt the judge model from existing LLMs, either closed-source (`o4-mini`, `Claude3.7`) or open-source (`DeepSeek-V3-0324`, `Qwen3-235B-A22B`, `QwQ-32B`). To enhance reliability and further reduce variance in the reward signal, we employ the following protocol: (i) alongside the candidate response, we provide the judge with a reference solution (e.g., a gold final answer, a brief solution outline, or a full reasoning trace when available); in our experiments, we draw this solution from a supervised fine-tuning dataset with correct answers and reasoning trajectories, which anchors the intended solution path and enables error localization; and (ii) we elicit step-wise evaluation rather than holistic evaluation. The judge model justifies correctness or flags an error sentence by sentence, identifies the first clear mistake, and then traces how this error propagates to the final incorrect conclusion.

Based on the reasoning trajectory score, we introduce a new outcome reward function:

$$r_{\texttt{SGPO}}(\mathbf{y}) = \begin{cases} 1, & \text{if the final answer of } \mathbf{y} \text{ is correct,} \\ \frac{1}{1+\exp(-\beta(\texttt{RTS}(\mathbf{y})-\gamma))}, & \text{otherwise.} \end{cases} \tag{2}$$

where $\gamma > 0$ and $\beta > 0$ (taken as 10 and 0.5 in the actual implementation) are two parameters to decide scale threshold and scale intensity, respectively. This design ensures that the model receives a more informative gradient signal during training, thereby encouraging refinement of partially correct reasoning rather than indiscriminate penalization of all incorrect outputs. This specification of $r_{\texttt{SGPO}}$ can be directly incorporated into the advantage calculation in Eq. (1). As a consequence, SGPO keeps the same rollout pipeline as GRPO and the same outcome-based supervision, and only replaces the reward used in within-group advantage computation from $r(x, y)$ to $r_{\texttt{SGPO}}(y)$ in Eq. (2).

**Remark 3.1.** *Our approach differs from process reward models (PRMs) (e.g. Lightman et al., 2024). For a prompt $\mathbf{x}$ and a prefix of reasoning steps $(a_1, \ldots, a_t)$, a PRM typically predicts either (i) a prefix-level value $V(\mathbf{x}, a_{1:t}) = \mathbb{P}(\text{final answer correct} \mid \mathbf{x}, a_{1:t})$, or (ii) a step-level progress signal such as $\Delta_t = V(\mathbf{x}, a_{1:t}) -$*

$V(\mathbf{x}, a_{1:t-1})$. *In practice, PRMs are trained by supervised ranking of intermediate steps and are used to re-rank trajectories or shape training at the* prefix *level, acting as approximate value (or Q-value) functions over prefixes. In contrast, SGPO introduces a different way of producing and using feedback signals: (i) Policy-guided rollouts without search. All trajectories are sampled from the current policy, without PRM-guided exploration or trajectory alteration; (ii) Post-hoc first-error identification. A step-wise judge inspects the entire trajectory, pinpoints the earliest error relative to a reference solution, and converts this into a calibrated scalar reward $r_{SGPO}(\mathbf{y})$ via the reasoning trajectory score; (iii) Stable credit assignment in all-negative-sample groups. By locating the first definitive mistake only after observing the full trace, SGPO eliminates the look-ahead ambiguity and feedback loops inherent to PRM-guided search (Zhang et al., 2024a), while avoiding the need for the judge to solve the problem or approximate a value function. We note that SGPO alleviates but does not fully eliminate degeneracy: if all trajectories in a group receive identical $r_{SGPO}$ (e.g., they fail at the same first-error position), the group-normalized advantages can still vanish, though this phenomenon is rarely observed in our experiments.*

**Remark 3.2.** *Our approach differs from knowledge distillation (e.g. Kang et al., 2023; Gu et al., 2024). The student model trained via knowledge distillation inherits the judge the model's failure, since it only imitates the judge model's outputs. In contrast, SGPO leverages the judge model to identify mistakes in the student's reasoning, providing learning signals that go beyond imitation and enabling improvements unattainable by knowledge distillation.*

## 3.2 Accelerating Learning Dynamics

We present a theoretical analysis to explain why SGPO outperforms GRPO. We study an idealized setting in which the step-wise judge provides accurate first-error localization. Even in this regime, establishing a general separation is technically subtle, so we focus on a stylized example. To this end, we consider the case when $H = 2$, where each step admits two possible actions $a_h \in 1, 2$ for $h = 1, 2$. Extending the analysis to general horizons or action spaces and imperfect (noisy or biased) judges is left to future work. This configuration follows prior works (Dayan, 1991; Li et al., 2024b), in which analogous examples were employed to validate theoretical insights. Without loss of generality, we assume a unique ground-truth response $\mathbf{y}_{\mathbf{x}}^\star = (2, 2)$ for the prompt $\mathbf{x}$. For clarity, we restrict the sample space to $(1, 1), (2, 1), (2, 2)$, excluding $(1, 2)$ since a correct reasoning step is unlikely to, and should not, follow an incorrect precursor.

To illustrate the effect of SGPO, we analyze the learning dynamics of SGPO and GRPO in this simplified setting. Under GRPO, the rewards are assigned as $r((2, 2)) = 1$ and $r((2, 1)) = r((1, 1)) = 0$, meaning that only selecting the "good" action 2 at both steps yields a positive reward. In contrast, SGPO assigns $r_{SGPO}((2, 2)) = 1$, $r_{SGPO}((2, 1)) = \frac{1}{2}$ and $r_{SGPO}((1, 1)) = 0$. The difference is that partial progress – choosing the "good" action 2 in the first step but failing at the second – receives no credit in GRPO yet proportional credit in SGPO. Here, $\frac{1}{2}$ is chosen for illustrative purposes to convey the qualitative behavior of the reward mechanism, while the exact values used in experiments are determined by Eq. (2).

The algorithm iteratively updates the policy parameter $\theta$ using samples drawn from the current policy $\pi_\theta$. We rewrite the generic GRPO update with a step size $\eta > 0$ as follows,

$$\theta^{(k+1)} = \theta^{(k)} + \eta \cdot g(\theta), \quad \text{where } g(\theta) = \frac{1}{NGH} \left( \sum_{i=1}^{N} \sum_{k=1}^{G} \sum_{h=1}^{H} s_\theta(\mathbf{x}^i, a_{1:h-1}^{i,k}) A_{i,k} \right),$$

where $N$ is the number of prompts, $G$ is the number of groups, $H$ is the number of reasoning steps in each response, $s_\theta(\mathbf{x}^i, a_{1:h-1}^{i,k}) := \nabla \theta \log \pi_\theta(a_t | \mathbf{x}, a_{1:h-1})$ is the score function, and the advantage $A_{i,k}$ is defined by Eq. (1) for each question $\mathbf{x}^i$. To distinguish, we denote $g_{GRPO}(\cdot)$ as the gradient estimator using classical outcome reward model $r$, and $g_{SGPO}(\cdot)$ as the gradient estimator using the reward $r_{SGPO}$ as proposed in Section 3.1.

In our analysis, we examine the population-level learning dynamics with $G = 2$, omitting clipping and importance sampling. In practice, importance sampling and clipping are important for training stability; we omit them here to simplify the analysis and leave their theoretical treatment to future work. For simplicity, we perform our analysis in the likelihood space rather than in the parameter space directly. Indeed, we

define the key quantities as follows

$$p \doteq \pi_{\theta_1}(a_1 = 2, |, \mathbf{x}) = \frac{e^{\theta_1^{\mathbf{x},2}}}{e^{\theta_1^{\mathbf{x},1}} + e^{\theta_1^{\mathbf{x},2}}}, \qquad q \doteq \pi_{\theta_2}(a_2 = 2, |, \mathbf{x}, a_1 = 2) = \frac{e^{\theta_2^{\mathbf{x},2,2}}}{e^{\theta_2^{\mathbf{x},2,1}} + e^{\theta_2^{\mathbf{x},2,2}}}.$$

Note that the original 4-dimensional parameter space defined by $\theta_1^{\mathbf{x},1}$, $\theta_1^{\mathbf{x},2}$, $\theta_2^{\mathbf{x},2,1}$ and $\theta_2^{\mathbf{x},2,2}$ in $\mathbb{R}$ is reduced to a 2-dimensional likelihood space defined by $p, q \in [0, 1]$.

For our simple stylized model, we compute the score functions in terms of likelihood parameters $p, q$ as follows,

$$s(a_1 = 1 \,|\, \mathbf{x}) = \begin{bmatrix} p \\ -p \\ 0 \\ 0 \end{bmatrix}, \quad s(a_1 = 2 \,|\, \mathbf{x}) = \begin{bmatrix} p - 1 \\ 1 - p \\ 0 \\ 0 \end{bmatrix},$$

and

$$s(a_2 = 1 \,|\, \mathbf{x}, a_1 = 2) = \begin{bmatrix} 0 \\ 0 \\ q \\ -q \end{bmatrix}, \quad s(a_2 = 2 \,|\, \mathbf{x}, a_1 = 2) = \begin{bmatrix} 0 \\ 0 \\ q - 1 \\ 1 - q \end{bmatrix}.$$

The responses can be drawn i.i.d. from the distribution as follows,

$$(a_1, a_2) = \begin{cases} (1,1), & \text{w.p. } 1 - p, \\ (2,1), & \text{w.p. } p(1 - q), \\ (2,2), & \text{w.p. } pq. \end{cases} .$$

The SGPO and GRPO training dynamics with population-level policy gradient can be computed exactly for the stylized model as follows,

$$\bar{g}_{\mathsf{SGPO}}(\theta) = \mathbb{E}[g_{\mathsf{SGPO}}(\theta)] = \frac{1}{2} \begin{bmatrix} p(p - 1) \\ p(1 - p) \\ p^2 q(q - 1) \\ p^2 q(1 - q) \end{bmatrix}, \quad \bar{g}_{\mathsf{GRPO}}(\theta) = \mathbb{E}[g_{\mathsf{GRPO}}(\theta)] = \frac{1}{2} \begin{bmatrix} p(p - 1)q \\ p(1 - p)q \\ pq(q - 1) \\ pq(1 - q) \end{bmatrix}.$$

Since $g_{\mathsf{GRPO}}(\theta)$ and $g_{\mathsf{SGPO}}(\theta)$ concentrate around $\bar{g}_{\mathsf{GRPO}}(\theta)$ and $\bar{g}_{\mathsf{SGPO}}(\theta)$ when the number of samples in each group is sufficiently large, it is reasonable to analyze the population-level dynamics for illustration. Note that the high-probability guarantees for the sample-level dynamics can be derived using concentration inequalities under certain conditions.

Denote $p_{\mathsf{GRPO}}^{(k)}$ and $q_{\mathsf{GRPO}}^{(k)}$ as the value of quantity $p$ and $q$ at iteration $k$ under GRPO. Analogously, $p_{\mathsf{SGPO}}^{(k)}$ and $q_{\mathsf{SGPO}}^{(k)}$ are the corresponding probabilities $p$ and $q$ at iteration $k$ under SGPO. We can explicitly write down the SGPO and GRPO update rules with $\eta = 1$ as follows,

$$\begin{cases} p_{\mathsf{SGPO}}^{(k+1)} = \exp(f_{11}(p_{\mathsf{SGPO}}^{(k)})), \\ q_{\mathsf{SGPO}}^{(k+1)} = \exp(f_{12}(p_{\mathsf{SGPO}}^{(k)}, q_{\mathsf{SGPO}}^{(k)})), \end{cases} \quad \text{and} \quad \begin{cases} p_{\mathsf{GRPO}}^{(k+1)} = \exp(f_{21}(p_{\mathsf{GRPO}}^{(k)}, q_{\mathsf{GRPO}}^{(k)})), \\ q_{\mathsf{GRPO}}^{(k+1)} = \exp(f_{22}(p_{\mathsf{GRPO}}^{(k)}, q_{\mathsf{GRPO}}^{(k)})), \end{cases}, \tag{3}$$

where the functions $f_{ij}$ are defined by

$$\begin{aligned} f_{11}(p) &= \log(p) + p(1 - p) - \log(1 - p + pe^{p(1-p)}), \\ f_{21}(p, q) &= \log(p) + p(1 - p)q - \log(1 - p + pe^{p(1-p)q}), \\ f_{12}(p, q) &= \log(q) + p^2 q(1 - q) - \log(1 - q + qe^{p^2 q(1-q)}), \\ f_{22}(p, q) &= \log(q) + pq(1 - q) - \log(1 - q + qe^{pq(1-q)}). \end{aligned} \tag{4}$$

Our theoretical findings are summarized in the following theorem.

**Theorem 3.3.** *Suppose that $p_{GRPO}^{(0)} = q_{GRPO}^{(0)} = p_{SGPO}^{(0)} = q_{SGPO}^{(0)} = \frac{1}{2}$ and $\eta = 1$[3] for GRPO and SGPO. Then, we have that (i) GRPO and SGPO achieve successful learning: $p_{GRPO}^{(k)}, q_{GRPO}^{(k)}, p_{SGPO}^{(k)}, q_{SGPO}^{(k)} \to 1$ as $k \to +\infty$; (ii) SGPO outperforms GRPO in learning the "good" action in the first step: $p_{SGPO}^{(k)} > p_{GRPO}^{(k)}$ for all $k \geq 1$; (iii) SGPO outperforms GRPO in learning the optimal policy: $p_{SGPO}^{(k)} q_{SGPO}^{(k)} > p_{GRPO}^{(k)} q_{GRPO}^{(k)}$ for all $k \geq 1$.*

*Proof.* To show (i), recall that the sequence $(p_{SGPO}^{(k)})_{k \in \mathbb{N}}$ is strictly increasing and bounded in $(0, 1)$ from Lemmas B.4(i) and B.4(ii), so it converges to some value $c \in (0, 1]$. Taking limit as $k \to \infty$:

$$1 = \lim_{k \to \infty} \frac{p_{SGPO}^{(k+1)}}{p_{SGPO}^{(k)}} = \lim_{k \to \infty} \frac{1}{(1 - p_{SGPO}^{(k)}) e^{-p_{SGPO}^{(k)}(1 - p_{SGPO}^{(k)})} + p_{SGPO}^{(k)}} = \frac{1}{(1-c)e^{-c(1-c)} + c}.$$

Using the simple Taylor lower bound $e^{-x} \geq 1 - x$, we have

$$1 = \frac{1}{(1-c)e^{-c(1-c)} + c} \geq \frac{1}{(1-c)(1-c(1-c)) + c} \implies (c-1)^2 \leq 0 \implies c = 1.$$

This shows $p_{SGPO}^{(k)} \to 1$ as $k \to \infty$. Similarly, we can show $q_{GRPO}^{(k)}, p_{SGPO}^{(k)}, q_{SGPO}^{(k)} \to 1$ as $k \to \infty$.

To show (ii), consider the base case:

$$
\begin{aligned}
p_{SGPO}^{(1)} &= \exp(f_{11}(p_{SGPO}^{(0)})) = \exp(\log p_{SGPO}^{(0)} + h_{p_{SGPO}^{(0)}}(p_{SGPO}^{(0)}(1 - p_{SGPO}^{(0)}))) \\
&= \exp(\log p_{GRPO}^{(0)} + h_{p_{GRPO}^{(0)}}(p_{GRPO}^{(0)}(1 - p_{GRPO}^{(0)}))) \\
&> \exp(\log p_{GRPO}^{(0)} + h_{p_{GRPO}^{(0)}}(p_{GRPO}^{(0)}(1 - p_{GRPO}^{(0)})q_{GRPO}^{(0)})) = \exp(f_{21}(p_{GRPO}^{(0)}, q_{GRPO}^{(0)})) = p_{GRPO}^{(1)},
\end{aligned}
$$

where the inequality follows from Lemma B.1(ii). Thus, we use induction and assume $p_{SGPO}^{(k)} > p_{GRPO}^{(k)}$ for some $k \geq 1$. Then, we have

$$
\begin{aligned}
p_{SGPO}^{(k+1)} &= \exp(f_{11}(p_{SGPO}^{(k)})) > \exp(f_{11}(p_{GRPO}^{(k)})) = \exp(\log p_{GRPO}^{(k)} + h_{p_{GRPO}^{(k)}}(p_{GRPO}^{(k)}(1 - p_{GRPO}^{(k)}))) \\
&> \exp(\log p_{GRPO}^{(k)} + h_{p_{GRPO}^{(k)}}(p_{GRPO}^{(k)}(1 - p_{GRPO}^{(k)})q_{GRPO}^{(k)})) = \exp(f_{21}(p_{GRPO}^{(k)}, q_{GRPO}^{(k)})) = p_{GRPO}^{(k+1)},
\end{aligned}
$$

where the first inequality uses Lemma B.1(i) and the second one uses Lemma B.1(ii). Thus, $p_{SGPO}^{(k+1)} > p_{GRPO}^{(k+1)}$ and the induction is done. We have proved that $p_{SGPO}^{(k)} > p_{GRPO}^{(k)}$ for all $k \geq 1$.

To show (iii), first notice that we can show $p_{GRPO}^{(k)} = q_{GRPO}^{(k)}$ for all $k \geq 0$ by induction. The base case is trivial by initialization. Suppose $p_{GRPO}^{(k)} = q_{GRPO}^{(k)}$ for some $k \geq 0$, then by noticing that $f_{21}(p, p) = f_{22}(p, p)$, we have

$$
\begin{aligned}
p_{GRPO}^{(k+1)} &= \exp(f_{21}(p_{GRPO}^{(k)}, q_{GRPO}^{(k)})) = \exp(f_{21}(p_{GRPO}^{(k)}, p_{GRPO}^{(k)})) \\
&= \exp(f_{22}(p_{GRPO}^{(k)}, p_{GRPO}^{(k)})) = \exp(f_{22}(p_{GRPO}^{(k)}, q_{GRPO}^{(k)})) = q_{GRPO}^{(k+1)}.
\end{aligned}
$$

Thus, by induction, $p_{GRPO}^{(k)} = q_{GRPO}^{(k)}$ for all $k \geq 0$. Now, we can reduce the update rule of $p_{GRPO}^{(k)}$ as

$$p_{GRPO}^{(k+1)} = \frac{1}{(1/p_{GRPO}^{(k)} - 1)\exp(-(p_{GRPO}^{(k)})^2(1 - p_{GRPO}^{(k)})) + 1}.$$

In addition, we recall the update rule of $p_{SGPO}^{(k)}$ and $q_{SGPO}^{(k)}$ as

$$
\begin{aligned}
p_{SGPO}^{(k+1)} &= \frac{1}{(1/p_{SGPO}^{(k)} - 1)\exp(-p_{SGPO}^{(k)}(1 - p_{SGPO}^{(k)})) + 1} \\
q_{SGPO}^{(k+1)} &= \frac{1}{(1/q_{SGPO}^{(k)} - 1)\exp(-(p_{SGPO}^{(k)})^2 q_{SGPO}^{(k)}(1 - q_{SGPO}^{(k)})) + 1}
\end{aligned}.
$$

It suffices to show $p_{SGPO}^{(k)} q_{SGPO}^{(k)} > (p_{GRPO}^{(k)})^2$ for all $k \geq 1$. We prove by induction. For the base case,

$$\sqrt{p_{SGPO}^{(1)} q_{SGPO}^{(1)}} = \sqrt{\frac{1}{1 + e^{-1/4}} \cdot \frac{1}{1 + e^{-1/16}}} > \frac{1}{1 + e^{-1/8}} = p_{GRPO}^{(1)}.$$

---

[3]We use the unit stepsize for simplicity. Our results are valid for any sufficiently small step size.

Table 1: Evaluation results on offline RL training. For each model, we report the baseline performance before RL training. We then report RL training results that uses only negative samples and positive samples, respectively. Performance across validation and training dataset (`LIMO`) is shown.

| | AMC23 avg@16 | AIME24 avg@16 | MATH500 pass@1 | Olympiads pass@1 | LIMO pass@1 |
|---|---|---|---|---|---|
| **Qwen2.5-14B-Instruct** | | | | | |
| Baseline | 58.59 | 14.58 | **80.40** | 41.78 | 31.70 |
| Negative Samples only | **61.88** | **15.21** | **80.40** | **42.37** | 30.11 |
| Positive Samples only | 61.72 | 14.58 | 79.80 | 42.07 | **38.68** |
| **Qwen2.5-32B-Instruct** | | | | | |
| Baseline | 64.22 | 17.08 | **83.60** | 45.93 | 34.64 |
| Negative Samples only | **69.53** | **20.42** | 83.00 | 46.37 | 36.47 |
| Positive Samples only | 66.87 | 18.75 | **83.60** | **47.41** | **41.86** |

The above inequality holds true since Lemma B.1(iv) implies

$$2\log(1 + e^{-1/8}) > \log(1 + e^{-1/4}) + \log(1 + e^{-1/16}),$$

It remains to show that $p_{\text{SGPO}}^{(k)} q_{\text{SGPO}}^{(k)} > (p_{\text{GRPO}}^{(k)})^2$ implies $p_{\text{SGPO}}^{(k+1)} q_{\text{SGPO}}^{(k+1)} > (p_{\text{GRPO}}^{(k+1)})^2$ for $k \geq 1$. By Lemma B.4(iii), we know $p_{\text{SGPO}}^{(k)} > q_{\text{SGPO}}^{(k)}$ for all $k \geq 1$. Thus, Lemma B.2 implies that

$$p_{\text{SGPO}}^{(k+1)} q_{\text{SGPO}}^{(k+1)} = \frac{1}{A(p_{\text{SGPO}}^{(k)})B(p_{\text{SGPO}}^{(k)}, q_{\text{SGPO}}^{(k)})} > \frac{1}{C\left(\sqrt{p_{\text{SGPO}}^{(k)} q_{\text{SGPO}}^{(k)}}\right)^2}.$$

Using Lemma B.1(iii), we complete the induction by applying our induction hypothesis:

$$\frac{1}{C\left(\sqrt{p_{\text{SGPO}}^{(k)} q_{\text{SGPO}}^{(k)}}\right)^2} = \left(\exp\left(f_{21}\left(\sqrt{p_{\text{SGPO}}^{(k)} q_{\text{SGPO}}^{(k)}}, \sqrt{p_{\text{SGPO}}^{(k)} q_{\text{SGPO}}^{(k)}}\right)\right)\right)^2 > \left(\exp(f_{21}(p_{\text{GRPO}}^{(k)}, p_{\text{GRPO}}^{(k)}))\right)^2 = (p_{\text{GRPO}}^{(k+1)})^2.$$

This completes the proof. $\square$

**Remark 3.4.** *Theorem 3.3 presents one of the first theoretical analyses of GRPO with multiple samples and multi-step reasoning in the context of LLM reasoning. The first part establishes that SGPO converges to the optimal policy. The second and third parts demonstrate that SGPO both accelerates the acquisition of partially correct reasoning steps and preserves partial reasoning ability even when the final answer is incorrect. Importantly, the theorem provides a **per-iteration** comparison of learning under different reward mechanisms – an aspect rarely examined in previous works. The provable improvement in learning the optimal policy is also consistent with our numerical findings. We plot the resulting learning curves of our numerical simulation in Figure 2. The left panel shows the probability of selecting the "good" action in the first step at iteration $k$ (i.e., $p_{SGPO}^{(k)}$ vs. $p_{GRPO}^{(k)}$), while the right panel shows the probability of learning the optimal policy (i.e., $p_{SGPO}^{(k)} q_{SGPO}^{(k)}$ vs. $p_{GRPO}^{(k)} q_{GRPO}^{(k)}$). The results align with the predictions of Theorem 3.3, demonstrating that the likelihood of learning the optimal policy under SGPO consistently exceeds that of GRPO across training.*

## 4 Experiments

We present the benefits of differentiating negative samples through experiments in both offline and online settings. Offline RL is more computationally efficient, offering faster training, reduced memory consumption, and improved stability. In contrast, online RL provides greater flexibility and learning capacity, and has become the standard approach in large-scale reasoning models such as DeepSeek-R1 (Guo et al., 2025a).

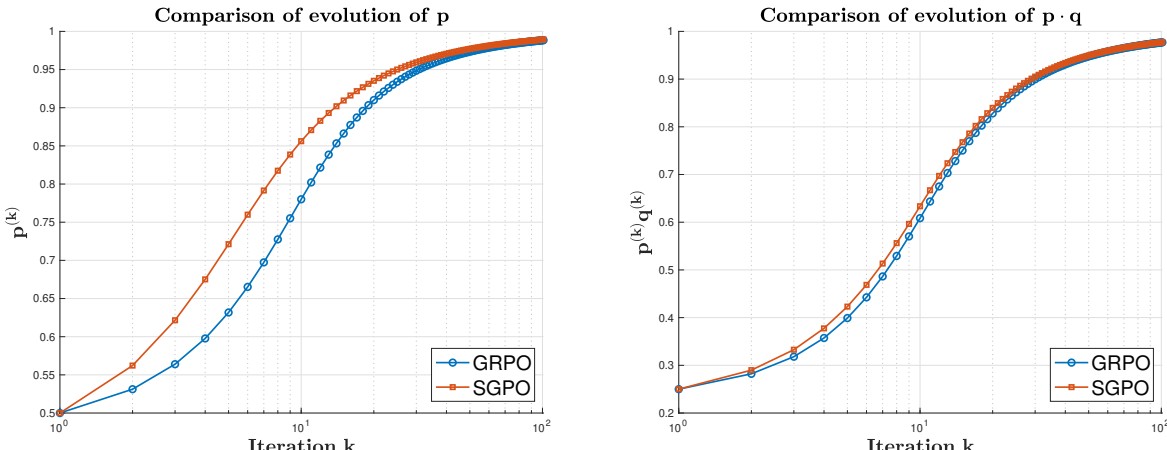

Figure 2: Learning dynamics of GRPO and SGPO in the simplified setting.

## 4.1 Offline Training

For baselines, we consider strong models without further fine-tuned on math-specific SFT datasets, namely `Qwen2.5-14B-Instruct` and `Qwen2.5-32B-Instruct`. Prior work showed that a small set of carefully curated prompts significantly enhance the reasoning capability. Accordingly, we adopt the `GAIR/LIMO` dataset (Ye et al., 2025) as the training set, which has demonstrated strong potential for improving the reasoning performance of large-scale (32B) models in offline SFT. Evaluation is conducted on four standard math reasoning benchmarks: `AIME24`, `AMC23`, `MATH500` (Hendrycks et al., 2021), and `OlympiadBench` (He et al., 2024). Our aim is to highlight the rich learning signal contained in all-negative-sample groups, showing that training exclusively on them can still yield performance gains. For benchmarks with fewer than 100 questions (`AMC23`, `AIME24`), we report `avg@16` results with a decoding temperature of 0.6 and $Top\_P = 0.95$. For benchmarks with more than 100 questions, we report `pass@1` results using greedy decoding. Here, pass@k is the percentage of prompts for which at least one of the $k$ sampled responses is correct, while avg@k is the average percentage of corrected samples among the $k$ samples. The maximum decoding length is set to 32768 tokens.

We conduct all response generation and model updates using offline RL (Peters & Schaal, 2007) with the standard GRPO mechanism. Specifically, the model is updated with advantages estimated from the offline dataset (see, e.g., Peng et al., 2019; Li et al., 2024b). For each prompt, we sample six responses per group and identify all-negative-sample groups in which all responses yield incorrect answers. Within these groups, we apply the step-wise judge model to assign differentiated rewards to negative samples, which are then used for offline RL updates. The model is trained for three epochs with a learning rate of $2 \times 10^{-6}$. As a contrastive baseline, we also perform offline RL using only positive rollouts with correct answers. This parallel setup enables a direct comparison between learning from exclusively negative reasoning trajectories and from exclusively positive ones.

We conduct offline RL training to demonstrate that utilizing all-negative-sample groups can enhance the reasoning abilities of LLMs. For comparison, we also include positive-only offline RL training. As shown in Table 1, SGPO with negative samples improves average performance and performs competitively on most benchmarks, and in some cases even surpasses models trained solely on positive samples. In particular, in the 14B model experiment, training on negative samples yields improvements on four benchmarks relative to the positive-sample baseline. These findings underscore the utility of negative samples, which should not be discarded in online GRPO training; see further comments in Section 4.4.

Table 2: Evaluation results on online RL training. We refer to BASELINE as the performance of the original model without RL finetuning. **Overall** is average performance across all the benchmarks. Note that the training dataset is `AIME1997-2023`. For `DeepSeek-R1-Distill-Qwen-7B`, we report additional results, including (i) compatibility with more judge models and (ii) ablation on the stability parameters $\beta$ and $\gamma$.

| | Kaoyan pass@1 | GradeMath pass@1 | MATH500 pass@1 | Olympiads pass@1 | CHMath24 avg@16 | AIME25 avg@16 | AIME24 avg@16 | GaoKao avg@16 | AMC23 avg@16 | Overall avg |
|---|---|---|---|---|---|---|---|---|---|---|
| **DeepSeek-R1-Distill-Qwen-7B** | | | | | | | | | | |
| BASELINE | 50.25 | 41.43 | 87.00 | 49.93 | 73.75 | **40.62** | 52.92 | 80.22 | 89.53 | 62.85 |
| GRPO | 55.78 | 43.33 | 89.40 | **56.00** | 71.04 | 36.68 | 52.08 | 80.30 | 88.91 | 63.72 |
| SGPO+o4-mini-0416 | 57.79 | 46.19 | 90.80 | 54.67 | 75.00 | 38.33 | 54.58 | 81.33 | 90.00 | 65.41 |
| SGPO+DeepSeek-V3-0324 | 54.77 | **47.17** | 91.00 | 55.11 | **77.29** | 40.42 | **56.87** | 82.28 | 90.83 | **66.19** |
| SGPO+Qwen3-235B-A22B | 56.78 | 46.67 | **92.00** | 54.67 | 73.33 | 37.92 | 55.63 | 81.17 | 90.63 | 65.42 |
| SGPO+QwQ-32B | 52.26 | 45.24 | **92.00** | 53.78 | 75.00 | 35.21 | 56.46 | **82.28** | 91.88 | 64.91 |
| SGPO+QwQ-32B w/o $\{\beta,\gamma\}$ | **58.29** | 42.38 | 90.20 | 55.11 | 74.58 | 38.69 | 53.63 | 81.24 | 88.75 | 65.08 |
| **DeepSeek-R1-Distill-Llama-8B** | | | | | | | | | | |
| BASELINE | 29.15 | 23.81 | 77.40 | 41.48 | **61.46** | 27.92 | **42.29** | 72.78 | 87.97 | 51.58 |
| GRPO | 35.68 | 28.33 | **84.00** | 46.32 | 57.08 | **28.33** | 42.08 | 68.99 | 86.72 | 53.06 |
| SGPO+Claude-3.7 | **39.70** | **29.05** | 83.60 | **48.44** | 58.96 | 24.58 | 39.37 | 71.52 | **89.06** | **53.81** |
| **Qwen2.5-14B-Instruct** | | | | | | | | | | |
| BASELINE | 37.69 | 49.52 | 80.40 | 41.78 | 21.88 | 13.13 | **14.58** | 41.14 | 58.59 | 39.85 |
| GRPO | **43.22** | 47.14 | 80.20 | 43.11 | 21.88 | 13.13 | 13.33 | 39.16 | **59.84** | 40.11 |
| SGPO+o4-mini-0416 | 38.69 | **53.33** | **81.00** | **44.00** | **22.92** | 16.67 | 14.17 | 39.00 | 59.22 | **41.00** |
| **Qwen2.5-32B-Instruct** | | | | | | | | | | |
| BASELINE | 45.73 | **53.81** | **83.60** | 45.93 | 26.87 | 12.29 | 17.08 | 44.15 | 64.22 | 43.74 |
| GRPO | **48.24** | 52.86 | 83.20 | 45.93 | 22.50 | 12.08 | **21.67** | **45.73** | 67.34 | 44.39 |
| SGPO+o4-mini-0416 | **48.24** | **53.81** | 83.00 | **46.81** | **29.79** | 14.58 | 19.58 | 45.09 | **69.53** | **45.06** |
| **QwQ-32B** | | | | | | | | | | |
| BASELINE | 64.32 | 62.38 | 94.60 | 68.74 | **89.39** | 68.54 | 77.71 | 86.88 | 97.03 | 78.84 |
| GRPO | 71.36 | 63.81 | 94.60 | 69.48 | 88.75 | 64.38 | 75.83 | **87.11** | 97.03 | 79.15 |
| SGPO+DeepSeek-V3-0324 | **73.37** | **64.76** | **95.00** | **70.22** | 88.33 | 66.46 | **78.33** | **87.11** | **97.97** | **80.17** |

## 4.2 Online Training

For baselines, we consider applying `Qwen2.5-14B-Instruct`, `Qwen2.5-32B-Instruct`, `QwQ-32B`, `DeepSeek-R1-Distill-Qwen-7B` and `DeepSeek-R1-Distill-Llama-8B`. Online GRPO training is implemented using the `verl` framework (Sheng et al., 2025). For the step-wise judge model, we adopt a diverse set of LLMs, ranging from closed-source models with strong reasoning capabilities (`o4-mini`, `Claude3.7`) to open-source models that are more accessible to the community, including `DeepSeek-V3-0324`, `Qwen3-235B-A22B`, and `QwQ-32B`.

Compared to offline RL, online RL yields larger improvements in a model's reasoning capabilities. Since our baselines already include strong distillation models, some benchmarks used in offline evaluation are nearing saturation. To provide a better assessment, we expand our evaluation suite beyond `AMC23`, `AIME24`, `MATH500`, and `OlympiadBench` by including `AIME25`, `GradeSchool` (Ye et al., 2025), `CHMath24`, `Kaoyan`, and `Gaokao`. Specifically, `CHMath24` is the benchmark from the 2024 Chinese High School Mathematics League Competition, `Gaokao` from China's 2024 National College Entrance Examination, `Kaoyan` from the Chinese Graduate School Entrance Examinations, and `GradeSchool` targets elementary-level mathematical reasoning. Among these, `CHMath24` and `Gaokao` each contain fewer than 100 questions, for which we apply the temperature-based decoding for evaluation.

For GRPO training, we use the `AIME` collections from 1997 to 2023 provided in DeepScaler (Luo et al., 2025b), training for 12 epochs. All training questions are in English, while evaluation benchmarks include multilingual questions. Notably, negative samples learned during training generalize well to out-of-domain mathematical reasoning tasks. SGPO training follows the same setup. With batch-simultaneous processing, judge model calls take 90 seconds per batch of negatives, adding 10% wall-clock time relative to rollout and update. Step-wise supervision is applied only to all-negative-sample groups during the first three epochs, as we expect this duration to suffice for the model to internalize corrective signals; beyond this point, unresolved examples are more indicative of model capacity limits than learnability. Accordingly, the end-to-end wall-

Table 3: Evaluation results are reported for `DeepSeek-R1-Distill-Qwen-7B` across four independent runs. First column indicates judge models and its corresponding reward stability setup.

| | Kaoyan pass@1 | GradeMath pass@1 | MATH500 pass@1 | Olympiads pass@1 | CHMath24 avg@16 | AIME25 avg@16 | AIME24 avg@16 | GaoKao avg@16 | AMC23 avg@16 | Overall avg |
|---|---|---|---|---|---|---|---|---|---|---|
| **DeepSeek-R1-Distill-Qwen-7B-SGPO** | | | | | | | | | | |
| +Qwen3-235B-A22B | $53.90 \pm 2.10$ | $46.55 \pm 0.24$ | $91.30 \pm 0.87$ | $53.45 \pm 1.35$ | $74.48 \pm 1.10$ | $37.40 \pm 1.05$ | $55.73 \pm 1.39$ | $81.33 \pm 0.18$ | $90.19 \pm 0.48$ | $64.92 \pm 0.37$ |
| +QwQ-32B | $53.89 \pm 1.66$ | $44.88 \pm 1.36$ | $91.15 \pm 0.84$ | $53.71 \pm 0.74$ | $74.33 \pm 1.29$ | $37.03 \pm 1.23$ | $54.76 \pm 1.87$ | $81.91 \pm 0.53$ | $90.08 \pm 1.23$ | $64.64 \pm 0.41$ |
| +QwQ-32B w/o $\{\beta,\gamma\}$ | $56.14 \pm 2.76$ | $44.53 \pm 2.41$ | $90.10 \pm 0.66$ | $53.64 \pm 1.08$ | $73.89 \pm 1.13$ | $38.70 \pm 1.80$ | $53.63 \pm 2.15$ | $81.24 \pm 0.50$ | $88.83 \pm 0.81$ | $64.52 \pm 0.57$ |

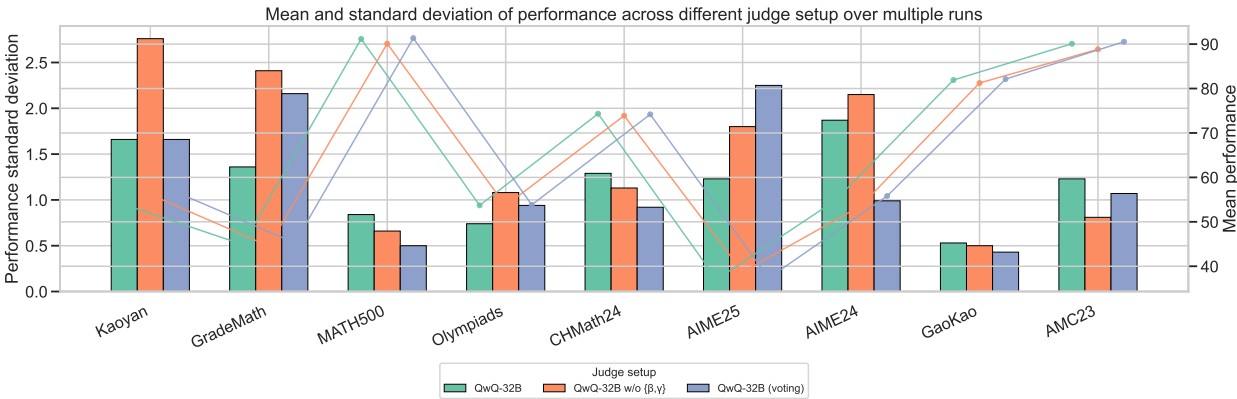

Figure 3: Mean and standard deviation over multiple runs across different judge model setups.

Table 4: Evaluation results are reported for `DeepSeek-R1-Distill-Qwen-7B` as base model and `QwQ-32B` as judge model with and without majority voting.

| | Kaoyan pass@1 | GradeMath pass@1 | MATH500 pass@1 | Olympiads pass@1 | CHMath24 avg@16 | AIME25 avg@16 | AIME24 avg@16 | GaoKao avg@16 | AMC23 avg@16 | Overall avg |
|---|---|---|---|---|---|---|---|---|---|---|
| **DeepSeek-R1-Distill-Qwen-7B-SGPO** | | | | | | | | | | |
| +QwQ-32B | $53.89 \pm 1.66$ | $44.88 \pm 1.36$ | $91.15 \pm 0.84$ | $53.71 \pm 0.74$ | $74.33 \pm 1.29$ | $37.03 \pm 1.23$ | $54.76 \pm 1.87$ | $81.91 \pm 0.53$ | $90.08 \pm 1.23$ | $64.64 \pm 0.41$ |
| +QwQ-32B with voting | $56.66 \pm 1.66$ | $45.24 \pm 2.16$ | $91.35 \pm 0.50$ | $53.82 \pm 0.94$ | $74.19 \pm 0.92$ | $37.35 \pm 2.25$ | $55.81 \pm 0.99$ | $82.12 \pm 0.43$ | $90.53 \pm 1.07$ | $65.23 \pm 0.18$ |

clock overhead is upper bounded by approximately $10\% \times 3/12 = 2.5\%$, and in practice this overhead can be offset by a small reduction of time needed to reach a target performance. For all models, rollout length is fixed at 8192 tokens and group size at 8. Models less than 8B are trained on 8 H100, 14B models on 16 H100, and 32B models on 32 H200. We adopt the default KL coefficient and learning rate from the `verl` training script (Sheng et al., 2025), and use the LIMO evaluation script (Ye et al., 2025), both of which are standard practices in the community. That being said, these benefits are not uniform across every benchmark. When negative samples are short, highly noisy, or dominated by early derailments, step-wise judging provides less actionable signal and can even introduce additional variance through judge noise, so SGPO may match or occasionally underperform GRPO or the baseline on specific tasks. This pattern is consistent with our empirical results and motivates a more nuanced view: SGPO is most effective when the model's failures retain informative intermediate structure (e.g., truncated near-correct trajectories or localized errors), rather than unstructured breakdowns.

A key insight from Table 2 is that stronger models generate higher-quality negative samples, which aid learning. As model capability improves, so does the informativeness of its mistakes. Negative samples fall into two categories: (i) correct reasoning trajectories truncated by output length limits, and (ii) trajectories containing logical errors. The first type remains highly valuable – yet discarded in GRPO – since it preserves meaningful reasoning steps, motivating our step-wise judge model. The second type, though incorrect, still provides informative signals, especially when all samples fail on genuinely challenging problems. Notably, stronger distilled models average 6K tokens per response, compared to only 1K tokens for weaker base models, making truncated but informative negative samples more common in the stronger case. Likewise, their erroneous responses tend to be richer and more useful for step-level judgment.

Table 5: Evaluation results are reported in terms of `pass@16` across benchmarks. The first two columns show the total number of questions and the number solved within 16 attempts, while the last two columns report the number of questions solved by SGPO but not by GRPO (SGPO\GRPO), and vice versa (GRPO\SGPO).

|          | SGPO - `pass@16` | GRPO - `pass@16` | SGPO \ GRPO | GRPO \ SGPO |
|----------|------------------|------------------|-------------|-------------|
| AIME24   | 23/30            | 19/30            | 4           | 0           |
| AIME25   | 21/30            | 21/30            | 1           | 1           |
| Gaokao   | 70/79            | 68/79            | 2           | 0           |
| AMC23    | 39/40            | 38/40            | 1           | 0           |
| CHMath24 | 27/30            | 25/30            | 2           | 0           |

### 4.3 Other Ablation Studies

To assess reliability of judge models, we evaluate our approach not only with strong closed-source reasoning models but also with publicly available models of weaker capacity: `DeepSeek-V3`, `Qwen3-235B` and `QwQ-32B`. As shown in Table 2 (best-tuned results) and Table 3 (multiple runs with weaker judges), performance remains stable, indicating that weaker judges do not significantly degrade outcomes. We attribute this reliability to two design choices: (i) first-step error identification with the reference answer. SGPO requires the judge only to verify each step against the reference, not to solve the problem, thereby reducing task difficulty and avoiding the pitfalls of generic PRMs; (ii) reward stability parameters $\beta$ and $\gamma$, which set the update inertia and reduce sensitivity to rewards from earlier failed rollouts. As confirmed by ablations, removing $\beta$ and $\gamma$ increases variance and weakens performance. To improve verification, we incorporate a Grok4−Heavy-inspired strategy: multiple independent evaluations by the judge model, with the error position selected by majority voting. Using `QwQ-32B` as the judge model, `DeepSeek-R1-Distill-Qwen-7B` as the base model, and four rollouts per judgment, we observed noticeable gains in consistency and stability (see Table 4). Figure 3 visualizes the aggregated results under different judge modes from Tables 3 and 4.

Although `avg@16` measures average performance across rollouts, `pass@16` reflects the ability to solve new questions with multiple attempts. As shown in Table 5, SGPO's gains in `pass@16` stem directly from leveraging negative samples. Learning only from solvable problems reinforces existing ability, whereas all-negative-sample groups correspond to genuinely difficult questions where GRPO consistently fails. These are precisely the cases where additional feedback can be most valuable. By providing step-level signals, SGPO rewards near-misses by reinforcing correct reasoning up to the first error, penalizes early failures by discouraging persistent error modes, and exposes blind spots by turning hard cases into informative training signals. In this regard, SGPO can provide additional benefits over GRPO, covering more hard problems and providing sharper credit assignment, which translates to more reliable learning under realistic compute budgets.

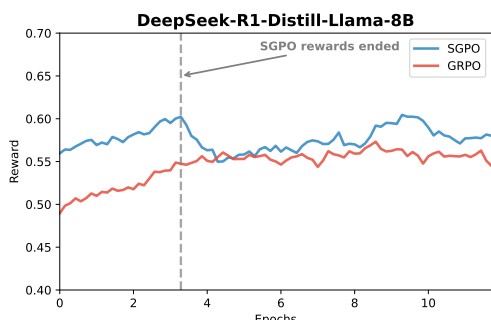

Figure 4: Evaluation results on GRPO and SGPO. SGPO rewards end at epoch 3.

By leveraging richer early-stage signals from negative samples, SGPO can achieve competitive performance with GRPO, and in some cases better performance or improved coverage on hard problems. As shown in Figure 4, SGPO continues improving beyond epoch 5 by solving several additional hard training problems, whereas GRPO plateaus. This improvement stems from informative negative samples that help resolve previously unsolved problems as also shown in Table 5. In line with our theoretical finding that SGPO converges faster than GRPO, empirical metrics offer supporting evidence. Prior work on RLVR entropy highlights its link to performance: Cui et al. (2025) showed that lower policy entropy under correct signals correlates with stronger policies, while Agarwal et al. (2025) demonstrated that directly minimizing entropy can improve performance. As shown in Figure 5, SGPO reduces policy entropy more rapidly than GRPO,

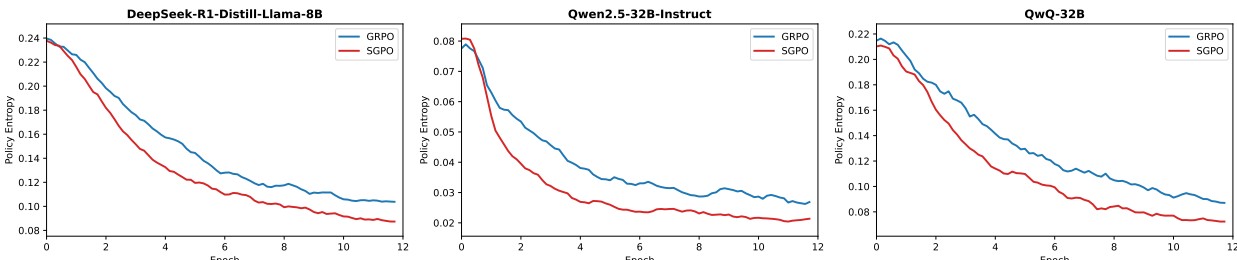

Figure 5: Policy entropy levels during training for GRPO and SGPO across different base models.

indicating faster convergence toward deterministic RLVR behavior with higher rollout confidence. This matches our theoretical results, confirming that step-wise signals accelerate convergence.

## 4.4 Discussions

We highlight the motivation for evaluating both offline and online RL. In the offline setup, training uses **only** negative samples, allowing us to directly test whether incorrect or incomplete reasoning trajectories can improve performance. In the online setup, we simulate realistic GRPO training, where batches contain a random mix of positive and negative samples. This demonstrates that negative samples are not only effective in isolation but also remain valuable in practical settings with noisier, mixed data. While mixing positives and negatives introduces noise, simply discarding negative samples does not stabilize training; in several cases, the performance of GRPO drops below baseline, as the models overfit to problems they can solve.

This instability arises from limited out-of-domain generalization and catastrophic forgetting. Without exposure to challenging or partially correct reasoning, the model risks overfitting to easy cases, reinforcing shallow heuristics instead of developing robust problem-solving skills. The absence of diverse failure cases can also cause catastrophic forgetting, degrading performance on previously solvable tasks. Incorporating negative samples mitigates these issues, and SGPO shows stronger robustness on the Chinese OOD math benchmarks we evaluate. We emphasize that SGPO does not guarantee uniform improvements on every benchmark: its gains depend on the structure of negative samples. SGPO is most beneficial when failures are due to truncated but largely correct trajectories or localized mistakes, whereas when negative samples fail very early or lack meaningful structure, the step-wise signal can be less informative and gains may diminish. These observations motivate further work on more stable training frameworks, including richer reward diversification mechanisms for handling negative samples and efficient RL methods beyond GRPO.

**Benchmarking scope.** Our goal is to isolate the benefit of learning from all-negative-sample groups in outcome-based, verifiable-reward post-training with group-relative updates. We therefore benchmark SGPO primarily against GRPO under matched training pipelines (same base models, data, rollout budget, group size, KL control, and optimizer settings), since SGPO is designed as a drop-in modification of GRPO's credit assignment only in all-negative groups. This choice makes the comparison controlled: improvements can be attributed to step-wise differentiation of negative samples rather than to changes in the broader RL recipe.

A natural question is how SGPO compares to methods that rely on process reward models (PRMs) or other step-level scoring mechanisms. Such comparisons would indeed further strengthen the empirical picture, but they introduce additional moving parts (PRM training data/labels, architecture, calibration, and inference overhead) that are orthogonal to our main question: given an outcome-verifiable setting and GRPO-style updates, can we recover learning signal from all-negative groups by cheaply localizing the first error? In this work, we keep the benchmark focused on GRPO and its outcome-based setting, and view PRM-based pipelines as complementary directions for future evaluation, especially when reliable PRMs and their training recipes are available and standardized. That said, we acknowledge that broader benchmarking (e.g., PRM-based pipelines) is an important next step, but given the recency and centrality of GRPO in outcome-based RLVR, a controlled GRPO-centered comparison is sufficient to establish relevance for the target setting.

## 5 Conclusion

We propose a simple and efficient framework that introduces response diversity within all-negative-sample groups and prove, in a simplified setting, that such diversification can accelerate the learning dynamic of GRPO. Empirically, SGPO improves average performance and improves coverage on harder problems, with the largest benefits in early and mid-training when all-negative groups are prevalent; gains are not uniform across benchmarks and depend on the structure/informativeness of negative samples. Future works include extending theoretical results to broader multi-step reasoning tasks, applying response diversity to accelerate other RL methods, and designing lightweight, task-specific reward models that evaluate reasoning steps correctly even if they cannot solve the full problem.

## Broader Impacts

Although large reasoning models can enable substantial social and economic benefits, a growing literature also highlights potential negative impacts, including environmental costs from training and inference, concentration of power and access, labor displacement, and downstream misuse (e.g., for manipulation or disinformation). Our work improves the post-training procedure for such models; as a result, it may contribute to both beneficial applications and accelerated adoption, which can amplify these broader concerns. We emphasize that SGPO is a technical contribution and does not address these systemic risks directly. Responsible deployment should consider energy and computing reporting, access and governance practices, and safeguards against misuse. More broadly, we view a rigorous assessment of societal impacts alongside technical progress as essential.

## Acknowledgment

We sincerely appreciate Buzz High Performance Computing (`https://www.buzzhpc.ai`, `info@buzzhpc.ai`) for providing computational resources and support for this work.

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

# A    Related Works

We comment on all related topics, including reasoning through chain-of-thought and its variants, test-time compute, direct preference alignment methods, reward models and reinforcement learning from AI feedback. For an overview of more reasoning models and methods, we refer to two recent surveys (Huang & Chang, 2023; Chen et al., 2025c).

**Chain-of-Thought and its variants.** Chain-of-thought (CoT) refers to as a broad class of methods that generate an intermediate reasoning process before arriving at a final answer. These approaches either prompt LLMs (Wei et al., 2022; Khot et al., 2023; Zhou et al., 2023) or train LLMs to generate reasoning chains through supervised fine-tuning (SFT) (Yue et al., 2024; Yu et al., 2024b; Li et al., 2025) and/or RL (Wang et al., 2024; Shao et al., 2024; Havrilla et al., 2024; Yu et al., 2025a). While CoT has proven effective for certain tasks, its auto-regressive generation nature makes it challenging to mimic human reasoning on more complex problems (LeCun, 2022; Hao et al., 2023), which require planning and search. Recent efforts were devoted to equipping LLMs with tree search methods (Xie et al., 2023; Yao et al., 2023; Hao et al., 2024a) or training LLMs on search trajectories (Lehnert et al., 2024; Gandhi et al., 2024; Su et al., 2025). Several other works have investigated why CoT is effective. For example, (Madaan et al., 2023) used a counterfactual prompting approach to examine the relative contributions of prompt elements, including symbols (digits, entities) and patterns (equations). (Feng et al., 2023; Merrill & Sabharwal, 2024; Li et al., 2024a) analyzed CoT from the perspective of model expressivity, and (Feng et al., 2023) showed that employing CoT increases the effective depth of a transformer since the generated outputs are looped back to the input. This insight motivated the chain-of-continuous-thought paradigm (Hao et al., 2024b), and a related approach has been proposed in (Cheng & Van Durme, 2024).

**Reasoning through test-time compute.** OpenAI-o1 (Jaech et al., 2024) is among the first large-scale applications of RL to reasoning, and achieved state-of-the-art performance upon release. Following this trend, DeepSeek-R1 (Guo et al., 2025a) is the first open-weight model to match or exceed OpenAI-o1. Their real-world success stories have involved several simple yet novel techniques that enhance LLM reasoning through more test-time compute, including chain-of-thought (Wei et al., 2022), self-consistency (Wang et al., 2023), best-of-$N$ sampling (Snell et al., 2025), process reward models (Lightman et al., 2024), exploration-exploitation mechanism (Chen et al., 2026b), Monte Carlo tree search (Silver et al., 2016; Hao et al., 2023), tree-of-thought (Yao et al., 2023), and recent works on preventing overthinking (Chen et al., 2024b; Team et al., 2025; Luo et al., 2025a; Arora & Zanette, 2025) and compressing chain-of-thought (Hao et al., 2024b; Cheng & Van Durme, 2024). More specifically, *chain-of-thought* is a reasoning approach where intermediate steps are explicitly written to make complex problem-solving processes more transparent and logical. *Self-consistency* suggests generating multiple final answers and returning the mode of an empirical distribution, enhancing test-time performance when test-time verifiers are unavailable. Unfortunately, it is computationally expensive and effective only when answers can be clustered. *Best-of-N sampling* resolves this issue by sampling answers from the model and selecting the best at test time according to the scoring function; however, it is sensitive to the accuracy of test-time scoring functions (Gao et al., 2023). *Process reward models* offer fine-grained supervision of chain-of-thought reasoning, but they might be vulnerable to reward hacking and introduce computation overhead. *Monte Carlo tree search* is a generic technique that allocates computational resources toward the most promising regions of the search space, and *tree-of-thought* and its extension (Besta et al., 2024; Gandhi et al., 2024) simplified this idea by exploring multiple reasoning paths in a specific structure, allowing language models to select the most promising line of thought for complex problem-solving. Both *length regularization* and *compressed chain-of-thought* are developed to reduce inference costs for reasoning, which is crucial for the economic feasibility, user experience and environmental sustainability of LLMs. In addition, several works have focused on specific reasoning tasks (Lampinen et al., 2024; Yang et al., 2025; Srivastava et al., 2024; Huang et al., 2025; 2024; Guo et al., 2025b; Gou et al., 2024; Wang et al., 2025; Guo et al., 2025c), demonstrating promising performance. The recent findings Xiong et al. (2025a) have shown that the REINFORCE-type methods (including GRPO (Shao et al., 2024)) can not effectively learn from all-negative-sample groups. Our work alleviates this issue by leveraging AI feedback to differentiate negative samples. We also provide a theoretical analysis through a stylized model, explaining why such diversification improves GRPO's learning dynamics.

**Direct preference alignment methods.** These methods (e.g., DPO (Rafailov et al., 2023)) are simple and stable offline alternatives to online RLHF. Various DPO variants with other objectives have been proposed, including ranking ones beyond pairwise preference data (Dong et al., 2023; Yuan et al., 2023a; Song et al., 2024; Chen et al., 2024a; Liu et al., 2025a) and simple ones that do not rely on a reference model (Hong et al., 2024; Meng et al., 2024). Since DPO does not train a reward model, the limited size of human labels becomes a bottleneck. To alleviate this limitation, subsequent works proposed to augment preference data using a trained SFT policy (Zhao et al., 2023) or a refined SFT policy with rejection sampling (Liu et al., 2024a). The DPO loss was recently rederived and extended to a token-level MDP view (Rafailov et al., 2024), where the state is the token prefix and the transition is deterministic – which has covered the fine-tuning of LLMs – and more general RL problems (Azar et al., 2024). There are other DPO variants (Ethayarajh et al., 2024; Park et al., 2024; Xu et al., 2024; Tang et al., 2024; Meng et al., 2024; Chen et al., 2025a; Zhao et al., 2025; Chen et al., 2026a). For example, Ethayarajh et al. (2024) designed a DPO-style loss variant using a prospect theory, Tang et al. (2024) optimized a general preference loss instead of the log-likelihood loss, and Meng et al. (2024) aligned the reward function in the preference optimization objective with the generation metric. Dong et al. (2024) and Xiong et al. (2024) proposed to generate human feedback in an online fashion to mitigate the distribution-shift and over-parameterization phenomenon. This improves DPO for complex reasoning tasks (Pang et al., 2024). Several other works focus on *unintentional alignment* of DPO and developing new methods (Pal et al., 2024; Tajwar et al., 2024; Liu et al., 2024b; Xiao et al., 2024; Yuan et al., 2025; Razin et al., 2025; Chen et al., 2025b). Among these works, Razin et al. (2025) proposed to measure the similarity between preferred and dispreferred responses using the centered hidden embedding similarity (CHES) score and showed that filtering out preference pairs with small CHES score improves DPO, while (Chen et al., 2025b) proposed to use comparison oracles, and showed that combining it with DPO effectively alleviated the issue of unintentional alignment.

**Reward models.** For the prompt $\mathbf{x}$ with a ground-truth response $\mathbf{y}_{\mathbf{x}}^{\star}$, we evaluate by implementing a regular expression match on the final answer (Hendrycks et al., 2021): $r(\mathbf{x}, \mathbf{y}) = 1$ if $\mathbf{y}$ matches $\mathbf{y}_{\mathbf{x}}^{\star}$ on the *final answer* and $r(\mathbf{x}, \mathbf{y}) = 0$ otherwise. An *outcome reward* model (ORM) (Cobbe et al., 2021; Uesato et al., 2022) is trained for estimating $r(\mathbf{x}, \mathbf{y})$. In particular, we first choose $\mathbf{x} \in \mathcal{D}$ and collect training samples $(\mathbf{x}, \mathbf{y} \sim \pi_{\theta}(\cdot|\mathbf{x}), r(\mathbf{x}, \mathbf{y}))$. Then, we take $(\mathbf{x}, \mathbf{y})$ as input and train an ORM to predict $r(\mathbf{x}, \mathbf{y})$. This can be done using binary classification (Cobbe et al., 2021; Yu et al., 2024a), direct preference optimization (Hosseini et al., 2024) or next-token prediction (Zhang et al., 2024b). Previous works also train LLMs on self-generated data using the ground-truth outcome reward model with either supervised fine-tuning (Singh et al., 2024; Yuan et al., 2023b; Zelikman et al., 2022) or online RL (Bi et al., 2024; Guo et al., 2025a). A *process reward* model (PRM) is trained to score $a_h$ at $\mathbf{s}_h = (\mathbf{x}, a_1, \ldots, a_{h-1})$ either using human annotations (Lightman et al., 2024) or the value functions based on LLM-generated data (Wang et al., 2024; Luo et al., 2024; Setlur et al., 2025); indeed, PRMs estimate either the likelihood of future success or the change in the likelihood of future success before and after taking $a_h$. In addition, PRMs were also developed to improve search methods (Snell et al., 2025; Wu et al., 2025), and to identify the "first pit" in an incorrect reasoning trajectory to construct preference pairs for direct preference alignment (Hwang et al., 2024; Setlur et al., 2024). Related work such as VinePPO (Kazemnejad et al., 2024) refines process-level credit assignment in PPO without explicitly training a PRM and avoids a learned value network by using Monte Carlo rollouts from intermediate prefixes; this is complementary to our GRPO-oriented setting.

**Reinforcement learning from AI feedback.** Reinforcement learning from human feedback (RLHF) uses human-preference-aligned reward models to evaluate response quality (Christiano et al., 2017; Ziegler et al., 2019; Stiennon et al., 2020; Ouyang et al., 2022). A key barrier to scale RLHF is the need for high-quality human labels. Previous studies (Gilardi et al., 2023; Ding et al., 2023) have shown that modern LLMs exhibit strong alignment with human judgments, suggesting that AI-generated labels can serve as a viable alternative. In this context, (Bai et al., 2022) was the first to explore RLAIF, jointly optimizing helpfulness and harmlessness using both human and AI-generated labels, and (Roit et al., 2023; Kwon et al., 2023; Lee et al., 2024) showed that LLMs can produce informative reward signals for RL post-training. Our approach can leverage AI feedback to introduce response diversity within all-negative-sample groups by assigning intermediate binary rewards to reasoning steps. Indeed, one identifies the proportion of correct steps in the reasoning trajectory and use it to compute a reward $r_i \in [0, 1)$.

## B  Missing Proofs

In this section, we present the derivations underlying the update rules in Section 3.2. In particular, we derive $g_{\mathrm{GRPO}}(\theta)$ and $g_{\mathrm{SGPO}}(\theta)$, and obtain the corresponding update forms in Eqs. (3) and (4).

To derive $g_{\mathrm{GRPO}}(\theta)$ and $g_{\mathrm{SGPO}}(\theta)$, we start by restating the score functions:

$$
s(a_1 = 1 \mid x) = \begin{bmatrix} p \\ -p \\ 0 \\ 0 \end{bmatrix}, \quad s(a_1 = 2 \mid x) = \begin{bmatrix} p-1 \\ 1-p \\ 0 \\ 0 \end{bmatrix},
$$

$$
s(a_2 = 1 \mid x, a_1 = 2) = \begin{bmatrix} 0 \\ 0 \\ q \\ -q \end{bmatrix}, \quad s(a_2 = 2 \mid x, a_1 = 2) = \begin{bmatrix} 0 \\ 0 \\ q-1 \\ 1-q \end{bmatrix}.
$$

Let $S(y) := \sum_{h=1}^{H} s(a_h \mid x, a_{1:h-1})$ denote the score-sum for trajectory $y = (a_1, a_2)$. Under the restricted trajectory space $\{(1,1), (2,1), (2,2)\}$, we have

$$
S(1,1) = \begin{bmatrix} p \\ -p \\ 0 \\ 0 \end{bmatrix}, \quad S(2,1) = \begin{bmatrix} p-1 \\ 1-p \\ q \\ -q \end{bmatrix}, \quad S(2,2) = \begin{bmatrix} p-1 \\ 1-p \\ q-1 \\ 1-q \end{bmatrix},
$$

and the trajectory probabilities are

$$
\mathbb{P}(1,1) = 1-p, \quad \mathbb{P}(2,1) = p(1-q), \quad \mathbb{P}(2,2) = pq.
$$

Recall $G = 2$ and $H = 2$. For a single prompt, our estimator averages over both samples and steps:

$$
g(\theta) = \frac{1}{GH} \left( \sum_{k=1}^{G} \sum_{h=1}^{H} s\left( a_h^{(k)} \mid x, a_{1:h-1}^{(k)} \right) A_k \right) = \frac{1}{GH} \left( \sum_{k=1}^{G} S(y^{(k)}) A_k \right),
$$

where $A_k$ is the within-group standardized advantage computed from the two rewards. For $G = 2$, whenever the two rewards are distinct, standardization yields $A_{\mathrm{high}} = +1$ and $A_{\mathrm{low}} = -1$; if the rewards are equal we set $A_1 = A_2 = 0$. Thus, for any ordered pair $(y^{(1)}, y^{(2)})$ with distinct rewards,

$$
g(\theta) = \frac{1}{GH} \left( S(y^{(1)}) - S(y^{(2)}) \right) \cdot \mathrm{sign}(r(y^{(1)}) - r(y^{(2)})).
$$

**GRPO.**  In GRPO, $r(2,2) = 1$ and $r(1,1) = r(2,1) = 0$. Thus, the only nonzero contributions come from mixed pairs $\{(2,2), (1,1)\}$ and $\{(2,2), (2,1)\}$. Using independence of the two samples and summing both orderings,

$$
\bar{g}_{\mathrm{GRPO}}(\theta) = \mathbb{E}[g(\theta)] = \frac{2}{GH} \left( \mathbb{P}(2,2)\mathbb{P}(1,1)(S(2,2) - S(1,1)) + \mathbb{P}(2,2)\mathbb{P}(2,1)(S(2,2) - S(2,1)) \right).
$$

With $G = H = 2$, $\frac{2}{GH} = \frac{1}{2}$. Substituting the probabilities and $S(\cdot)$ yields

$$
\bar{g}_{\mathrm{GRPO}}(\theta) = \frac{1}{2} \begin{bmatrix} p(p-1)q \\ p(1-p)q \\ pq(q-1) \\ pq(1-q) \end{bmatrix}.
$$

**SGPO.**  In SGPO, $r_{\mathrm{SGPO}}(2,2) = 1$, $r_{\mathrm{SGPO}}(2,1) = \frac{1}{2}$, and $r_{\mathrm{SGPO}}(1,1) = 0$. Therefore, all three unordered mixed pairs contribute: $\{(2,2), (2,1)\}$, $\{(2,2), (1,1)\}$, and $\{(2,1), (1,1)\}$. Summing both orderings, we have

$$
\bar{g}_{\mathrm{SGPO}}(\theta) = \frac{2}{GH} \left( \mathbb{P}(2,2)\mathbb{P}(2,1)(S(2,2) - S(2,1)) + \mathbb{P}(2,2)\mathbb{P}(1,1)(S(2,2) - S(1,1)) + \mathbb{P}(2,1)\mathbb{P}(1,1)(S(2,1) - S(1,1)) \right).
$$

Again, we have $\frac{2}{GH} = \frac{1}{2}$ for $G = H = 2$, and substituting gives

$$\bar{g}_{\text{SGPO}}(\theta) = \frac{1}{2} \begin{bmatrix} p(p-1) \\ p(1-p) \\ p^2 q(q-1) \\ p^2 q(1-q) \end{bmatrix}.$$

Now, we derive the update rules in Eq. (3) and the functions in Eq. (4) from the population gradients. Let $\theta^{(k+1)} = \theta^{(k)} + \eta \bar{g}(\theta^{(k)})$, where $\bar{g}$ is either $\bar{g}_{\text{SGPO}}$ or $\bar{g}_{\text{GRPO}}$. For notational brevity, define the step-1 logits $(\theta_1, \theta_2) = (\theta^1_{x,1}, \theta^1_{x,2})$ and step-2 logits $(\theta_3, \theta_4) = (\theta^2_{x,2,1}, \theta^2_{x,2,2})$. In addition, we have

$$p = \frac{e^{\theta_2}}{e^{\theta_1} + e^{\theta_2}}, \quad q = \frac{e^{\theta_4}}{e^{\theta_3} + e^{\theta_4}}.$$

**Update of $p$.** Let $g_1, g_2$ denote the first two coordinates of $\bar{g}(\theta)$. Then, we have

$$p^{(k+1)} = \frac{e^{\theta_2^{(k)} + \eta g_2}}{e^{\theta_1^{(k)} + \eta g_1} + e^{\theta_2^{(k)} + \eta g_2}} = \frac{e^{\theta_2^{(k)}} e^{\eta g_2}}{e^{\theta_1^{(k)}} e^{\eta g_1} + e^{\theta_2^{(k)}} e^{\eta g_2}}.$$

Using $p^{(k)} = \frac{e^{\theta_2^{(k)}}}{e^{\theta_1^{(k)}} + e^{\theta_2^{(k)}}}$ and $1 - p^{(k)} = \frac{e^{\theta_1^{(k)}}}{e^{\theta_1^{(k)}} + e^{\theta_2^{(k)}}}$, we can factor out $e^{\theta_1^{(k)}} + e^{\theta_2^{(k)}}$ to obtain

$$p^{(k+1)} = \frac{p^{(k)} e^{\eta(g_2 - g_1)}}{(1 - p^{(k)}) + p^{(k)} e^{\eta(g_2 - g_1)}}.$$

For SGPO, from $\bar{g}_{\text{SGPO}}(\theta)$ we have $g_1 = \frac{1}{2} p(p-1)$ and $g_2 = \frac{1}{2} p(1-p)$, hence $g_2 - g_1 = p(1-p)$. For GRPO, from $\bar{g}_{\text{GRPO}}(\theta)$ we have $g_1 = \frac{1}{2} p(p-1)q$ and $g_2 = \frac{1}{2} p(1-p)q$, hence $g_2 - g_1 = p(1-p)q$. Setting $\eta = 1$ gives the claimed update

$$p^{(k+1)}_{\text{SGPO}} = \frac{p^{(k)}_{\text{SGPO}} e^{p^{(k)}_{\text{SGPO}} (1 - p^{(k)}_{\text{SGPO}})}}{1 - p^{(k)}_{\text{SGPO}} + p^{(k)}_{\text{SGPO}} e^{p^{(k)}_{\text{SGPO}} (1 - p^{(k)}_{\text{SGPO}})}} = \exp(f_{11}(p^{(k)}_{\text{SGPO}})),$$

$$p^{(k+1)}_{\text{GRPO}} = \frac{p^{(k)}_{\text{GRPO}} e^{p^{(k)}_{\text{GRPO}} (1 - p^{(k)}_{\text{GRPO}}) q^{(k)}_{\text{GRPO}}}}{1 - p^{(k)}_{\text{GRPO}} + p^{(k)}_{\text{GRPO}} e^{p^{(k)}_{\text{GRPO}} (1 - p^{(k)}_{\text{GRPO}}) q^{(k)}_{\text{GRPO}}}} = \exp(f_{21}(p^{(k)}_{\text{GRPO}}, q^{(k)}_{\text{GRPO}})).$$

Taking log on both sides yields

$$f_{11}(p) = \log p + p(1-p) - \log\left(1 - p + p e^{p(1-p)}\right),$$

$$f_{21}(p, q) = \log p + p(1-p)q - \log\left(1 - p + p e^{p(1-p)q}\right).$$

**Update of $q$.** An identical argument applies to $q$ using the last two coordinates $g_3, g_4$ of $\bar{g}(\theta)$:

$$q^{(k+1)} = \frac{q^{(k)} e^{\eta(g_4 - g_3)}}{(1 - q^{(k)}) + q^{(k)} e^{\eta(g_4 - g_3)}}.$$

For SGPO, $g_3 = \frac{1}{2} p^2 q(q-1)$ and $g_4 = \frac{1}{2} p^2 q(1-q)$, so $g_4 - g_3 = p^2 q(1-q)$. For GRPO, $g_3 = \frac{1}{2} pq(q-1)$ and $g_4 = \frac{1}{2} pq(1-q)$, so $g_4 - g_3 = pq(1-q)$. With $\eta = 1$ this gives

$$q^{(k+1)}_{\text{SGPO}} = \exp(f_{12}(p^{(k)}_{\text{SGPO}}, q^{(k)}_{\text{SGPO}})), \quad q^{(k+1)}_{\text{GRPO}} = \exp(f_{22}(p^{(k)}_{\text{GRPO}}, q^{(k)}_{\text{GRPO}})),$$

where by taking log on both sides, we have

$$f_{12}(p, q) = \log q + p^2 q(1-q) - \log\left(1 - q + q e^{p^2 q(1-q)}\right),$$

$$f_{22}(p, q) = \log q + pq(1-q) - \log\left(1 - q + q e^{pq(1-q)}\right).$$

We then provide several technical lemmas that are important to the proof of Theorem 3.3. Indeed, the first lemma summarizes the properties of particular functions related to the aforementioned functions $f_{11}$, $f_{21}$, $f_{12}$ and $f_{22}$ from Eq. (4).

**Lemma B.1.** *The following statements hold true,*

*(i) The function $f_{11}$ is strictly increasing on $(0, 1)$.*
*(ii) The function $h_p(x) := x - \log(1 - p + pe^x)$ is strictly increasing for any fixed $p \in (0, 1)$.*
*(iii) The function $f_{21}$ is strictly increasing in either $p$ for any fixed $q$ or $q$ for any fixed $p$ on $(0, 1)$.*
*(iv) The function $\varphi(x) := \log(1 + e^{-e^x})$ is strictly concave on $(-\infty, 0)$.*

*Proof.* First of all, we have

$$f'_{11}(p) = \frac{1 + (1 - 2p)p(1 - p)}{p(1 - p + pe^{p(1-p)})} > \frac{3}{4p(1 - p + pe^{p(1-p)})} > 0.$$

Thus, the function $f_{11}$ is strictly increasing on $(0, 1)$.

Furthermore, we have

$$h'_p(x) = 1 - \frac{pe^x}{1 - p + pe^x} = \frac{1 - p}{1 - p + pe^x} \overset{0 < p < 1}{>} 0.$$

Thus, the function $h_p(x)$ is strictly increasing.

Moreover, we have

$$\frac{\partial f_{21}(p, q)}{\partial p} = \frac{1 + q(1 - 2p)p(1 - p)}{p(1 - p + pe^{qp(1-p)})} > \frac{3}{4p(1 - p + pe^{p(1-p)})} > 0,$$

$$\frac{\partial f_{21}(p, q)}{\partial q} = \frac{p(1 - p)^2}{1 - p + pe^{qp(1-p)}} > 0.$$

Thus, the function $f_{21}$ is strictly increasing in either $p$ for any fixed $q$ or $q$ for any fixed $p$ on $(0, 1)$.

Finally, we have

$$\varphi''(x) = \frac{(e^{x + e^x} - e^{e^x} - 1)e^x}{e^{2e^x} + 2e^{e^x} + 1}.$$

Since $u = e^x \in (0, 1)$ for $x < 0$, we have

$$(ue^u - e^u - 1)u = (e^u(u - 1) - 1)u < -u < 0.$$

Thus, $\varphi''(x) < 0$ for all $x < 0$ which shows that $\varphi$ is strictly concave on $(-\infty, 0)$. $\qquad\square$

The second lemma presents an inequality which plays a key role in the proof of Theorem 3.3.

**Lemma B.2.** *We define the auxiliary functions as follows,*

$$A(x) = 1 + \left(\tfrac{1}{x} - 1\right)e^{-x(1-x)}, \quad B(x, y) = 1 + \left(\tfrac{1}{y} - 1\right)e^{-x^2 y(1-y)},$$

$$C(z) = 1 + \left(\tfrac{1}{z} - 1\right)e^{-z^2(1-z)}.$$

*Then, we have $C(\sqrt{xy})^2 > A(x)B(x, y)$ for all $x$ and $y$ satisfying $1/2 < y < x < 1$.*

*Proof.* We consider the lower and upper bound of $e^{-u}$ when $u > 0$:

$$1 - u + \tfrac{u^2}{2} - \tfrac{u^3}{6} < e^{-u} < 1 - u + \tfrac{u^2}{2}.$$

Since $1/x - 1$, $1/y - 1$ and $1/\sqrt{xy} - 1$ are all positive, we have

$$A(x) \leq 1 + \tfrac{1-x}{x}\left(1 - x(1 - x) + \tfrac{x^2(1-x)^2}{2}\right) = \tfrac{1}{x} - (1 - x)^2 + \tfrac{x(1-x)^3}{2}.$$

$$B(x, y) \leq 1 + \tfrac{1-y}{y}\left(1 - x^2 y(1 - y) + \tfrac{x^4 y^2(1-y)^2}{2}\right) = \tfrac{1}{y} - x^2(1 - y)^2 + \tfrac{x^4 y(1-y)^3}{2}.$$

$$C(z) \geq 1 + \tfrac{1-z}{z}\left(1 - z^2(1 - z) + \tfrac{z^4(1-z)^2}{2} - \tfrac{z^6(1-z)^3}{6}\right)$$
$$= \tfrac{1}{z} - z(1 - z)^2 + \tfrac{z^3(1-z)^3}{2} - \tfrac{z^5(1-z)^4}{6}.$$

Set $z^2 = xy$, the original statement is equivalent to $(zC(z))^2 > (xA(x))(yB(x,y))$. Using the above upper and lower bound, it suffices to show $C_1(\sqrt{xy})^2 > A_1(x)B_1(x,y)$ where

$$A_1(x) = 1 - x(1-x)^2 + x^2(1-x)^3/2, \ B_1(x,y) = 1 - x^2y(1-y)^2 + x^4y^2(1-y)^3/2,$$
$$C_1(z) = 1 - z^2(1-z)^2 + z^4(1-z)^3/2 - z^6(1-z)^4/6.$$

By Lemma B.3, this is indeed true. This completes the proof. $\qquad\square$

**Lemma B.3.** *Define functions*

$$A_1(x) = 1 - x(1-x)^2 + x^2(1-x)^3/2, \ B_1(x,y) = 1 - x^2y(1-y)^2 + x^4y^2(1-y)^3/2,$$
$$C_1(z) = 1 - z^2(1-z)^2 + z^4(1-z)^3/2 - z^6(1-z)^4/6.$$

*Then, $C_1(\sqrt{xy})^2 > A_1(x)B_1(x,y)$ for all $1/2 < y < x < 1$.*

*Proof.* Let $x = u^2$ and $y = v^2$, then $1 > u > v > 1/\sqrt{2}$ and $z = uv$. We next show the desired inequality holds on a larger region, i.e., $1 > u > v > 2/3$. On this larger region, we have the reparameterization as follows:

$$u = \tfrac{2s+3}{3s+3}, \quad v = \tfrac{2r+2s+3}{3r+3s+3}, \quad s, r \in (0, +\infty),$$

or equivalently,

$$s = \tfrac{3(1-u)}{3u-2}, \quad r = \tfrac{3(u-v)}{(3u-2)(3v-2)}, \quad 1 > u > v > \tfrac{2}{3}.$$

It is easy to see this defines a one-to-one correspondence from $(u,v)$-space to $(s,r)$-space. Thus, we aim to prove the following function $f$ is positive:

$$F(s,r) := C_1\left(\tfrac{2s+3}{3s+3} \cdot \tfrac{2r+2s+3}{3r+3s+3}\right)^2 - A_1\left(\left(\tfrac{2s+3}{3s+3}\right)^2\right)B_1\left(\left(\tfrac{2s+3}{3s+3}\right)^2, \left(\tfrac{2r+2s+3}{3r+3s+3}\right)^2\right).$$

By leveraging Sympy's symbolic engine, the function expands and simplifies to:

$$F(s,r) = \tfrac{f(s,r)}{c(s+1)^{20}(r+s+1)^{20}}, \text{ where } f(s,r) := \sum_{k=0}^{20} c_{20-k}(s)r^{20-k},$$

where $c > 0$ is a universal constant, and single-variable polynomials $c_{20}(s), \ldots, c_2(s), c_0(s) > 0$ and $\Delta_2 := c_1(s)^2 - 4c_2(s)c_0(s) < 0$, for all $s > 0$ (see Table 6 for details). Notice that from the table, we can see the only nontrivial parts are $c_3(s) > 0$ and $c_2(s) > 0$ because only these two contain negative coefficients. The positivity of $c_3(s)$ is simple because there is only one term ($s^9$) with negative coefficient and for all $s > 0$,

$$19471456710454363005152664s^{10} + 9684588377731643071927236s^8 > 14413823109350224541499726s^9.$$

To see this, simple estimation and AM-GM inequality yield

$$\text{LHS} > 1.9 \times 10^{25}s^{10} + 9.6 \times 10^{24}s^8 > 2\sqrt{182.4} \times 10^{24}s^9 > 2.7 \times 10^{25}s^9 > \text{RHS}.$$

The positivity of $c_2(s)$ is more complicated because it has 4 negative terms $s^{10}, s^9, s^8, s^7$. However, we can use similar idea, i.e., choosing a pair of positive terms to bound a negative term:

$$95791062786555508724088742320s^{15} + 571809550541807937530952s^5 > 7027359523643236832970716s^{10};$$
$$437858623301624990522209529768s^{14} + 18478934378953446115053 0s^4 > 99854072704322871537392604s^9;$$
$$16326736853527122991715155824s^{13} + 3548837556962247216924 0s^3 > 3572603137796979208818 8925s^8;$$
$$46082190500843267907489331 53s^{12} + 4362950858813170449228s^2 > 60990378953076701422876 08s^7.$$

To see this, simple estimation and AM-GM inequality yield

$$\text{LHS} > 9.5 \times 10^{28}s^{15} + 5.7 \times 10^{23}s^5 > 2\sqrt{541.5} \times 10^{25}s^{10} > 4.6 \times 10^{26}s^{10} > \text{RHS};$$
$$\text{LHS} > 4.3 \times 10^{28}s^{14} + 1.8 \times 10^{23}s^4 > 2\sqrt{77.4} \times 10^{25}s^9 > 1.7 \times 10^{26}s^{10} > \text{RHS};$$
$$\text{LHS} > 1.6 \times 10^{28}s^{13} + 3.5 \times 10^{22}s^3 > 2\sqrt{5.6} \times 10^{25}s^9 > 4.7 \times 10^{25}s^8 > \text{RHS};$$
$$\text{LHS} > 4.6 \times 10^{27}s^{12} + 4.3 \times 10^{21}s^2 > 2\sqrt{19.7} \times 10^{24}s^7 > 8.8 \times 10^{24}s^7 > \text{RHS}.$$

In conclusion, we have all coefficient $c_i(s)$ positive except $c_1(s)$, but it doesn't affect the positivity of $f$ because $\Delta_2 < 0$. This completes the proof. □

Table 6: Coefficient Lists of $F(s, r)$

| Notation | Value |
|---|---|
| $c$ | 4376759565260494436836 |
| $c_{20}$ | $2(2s+3)^2(1714774320744848750s^{18} + 26610409260691576200s^{17} + 191778746468802317181s^{16} + 850194149855082319224s^{15} + 2587082434290045806049s^{14} + 5704415906039160731874s^{13} + 9366197581963232054460s^{12} + 11563054951307567026248s^{11} + 10670965452123886149660s^{10} + 7187176769582075261292s^9 + 3372972168996579430017s^8 + 1072082836158220703952s^7 + 370302094042890771285s^6 + 32990167737690242581 8s^5 + 282799986805616267862s^4 + 151168270170893365008s^3 + 49139849518345513368s^2 + 9090603727935062976s + 742484948385838248)$ |
| $c_{19}$ | $8(2s+3)^2(8573871603724243750s^{19} + 141183751848798840450s^{18} + 1085065457535611084097s^{17} + 5159905424662678527663s^{16} + 16962017821014041355285s^{15} + 40761515892906930261393s^{14} + 73777358861762677983126s^{13} + 101968163476291942277643s^{12} + 107719550183279202945336s^{11} + 85951871005332247942347s^{10} + 50477142420763872747039s^9 + 21228461710484227270812s^8 + 7175576253360286202193s^7 + 3821642119447908810138s^6 + 3343140602539615070982s^5 + 2308058741380310946144s^4 + 1046528691565621471344s^3 + 300769285860744146028s^2 + 50403014643440592936s + 3785769852305984190)$ |
| $c_{18}$ | $2(2s + 3)^2(32580712094152126250 0s^{20} + 5673987380977312396200s^{19} + 46320143614200183575358s^{18} + 235164624868455434314740s^{17} + 830330782346499878631402s^{16} + 2159105892766696625508432s^{15} + 4268066364777710628878112s^{14} + 6521177726276586191628264s^{13} + 7742933419720025359660131s^{12} + 7110911361873582109051992s^{11} + 4976474876383070663195517s^{10} + 2600230719135591148269222s^9 + 10353079281161503861096 95s^8 + 420495220158165783672300s^7 + 287436345862168026209421s^6 + 225749912117527047120354s^5 + 1322055539247657570232 86s^4 + 52546262098747895532864s^3 + 13596497258861930544108s^2 + 2087305777245729888936s + 145293329180967197454)$ |
| $c_{17}$ | $12(2s + 3)^2(32580712094152126250 0s^{21} + 5982992191700268855300s^{20} + 51700930512613327403406s^{19} + 279079780777259477944590s^{18} + 1053187317945045364767966s^{17} + 2945458902809849586876810s^{16} + 6310900331865363012743844s^{15} + 10554182290179327214273482s^{14} + 13894164004300292404029789s^{13} + 14397511755502695216399777s^{12} + 11648187532971253859583195s^{11} + 7253672843249894875203621s^{10} + 3463934076062711984148183s^9 + 14013118711246369225106 79s^8 + 70963507345380338433066 3s^7 + 526601300781722718969621s^6 + 366411462920903557014120s^5 + 187790938366917450633606s^4 + 66828265788979238418684s^3 + 15774993964318985462592s^2 + 2238177282159012945966s + 145311275743961970078)$ |
| $c_{16}$ | $3(2s + 3)^2(5538721056005861462500s^{22} + 10696394904119483034480 0s^{21} + 975372417387075095557110s^{20} + 5577701869567635624516312s^{19} + 22401070138854784562671602s^{18} + 6703472556180932146225722 0s^{17} + 154689592660061775780683034s^{16} + 280889539801332767094091608s^{15} + 405664328993005936098158220s^{14} + 467432234597450323377654624s^{13} + 428162911701836470453488816s^{12} + 308851193177419859648120664s^{11} + 1739237115076245954125557 92s^{10} + 78611043902295277305014364s^9 + 345027239321086599680681 91s^8 + 20784869678087847011350512s^7 + 152110704912752708992604 79s^6 + 94397951694788177205573 90s^5 + 43236427772992108674668 40s^4 + 13983362552252252566868617 6s^3 + 3041651033834191453666 80s^2 + 401701709844688883968 80s + 2445688799534592091926)$ |

| | |
|---|---|
| $c_{15}$ | $12(2s + 3)^2(4430976844804689170000s^{23} + 89773624658788072119600s^{22} + 86144978996747896790296 8s^{21} + 520205234057036396179927 2s^{20} + 2215095867472214763688788 0s^{19} + 706107689774315971385024 16s^{18} + 17454774771903874157924914 8s^{17} + 34184679883022075974400680 2s^{16} + 53701789749405047322088747 5s^{15} + 68038631708864390907265235 7s^{14} + 69490091744552770910260620 3s^{13} + 56885429435145235255915535 7s^{12} + 37016250086232520818694940 7s^{11} + 19208993461527481014311363 7s^{10} + 855600003744048710780447 43s^9 + 42188979069325595690484894s^8 + 2798648298363544891614947 7s^7 + 1934947300052794842316009 8s^6 + 1082654952241765058234790 3s^5 + 449911900941991185014790 3s^4 + 1337268081929646109116429s^3 + 270188727447397870150299s^2 + 334121914487222028717 93s + 1916481339467789227047)$ |
| $c_{14}$ | $3(2s + 3)^2(4430976844804689170000 0s^{24} + 9397609008462027996336 00s^{23} + 946588223144997927058161 6s^{22} + 601894156442872654302402 24s^{21} + 27083185113636696116985912 0s^{20} + 9160822434227131933406504 64s^{19} + 2414605718335618880853970 032s^{18} + 50717848050504100358231675 76s^{17} + 8606003351786628019139811 024s^{16} + 11882072721559268002578594 336s^{15} + 133731357360665889152672251 92s^{14} + 12233789753017327381398656 592s^{13} + 9038087243602138768195078 704s^{12} + 53692865526908734462761173 28s^{11} + 262353585257354926709155530 0s^{10} + 11960717809829396518651440 96s^9 + 6600782446234967802635880 75s^8 + 4513931959976662086674588 52s^7 + 2903436770732689418640463 05s^6 + 14804204283262980396732105 0s^5 + 5646419333104340411674178 4s^4 + 1555947843019285082981882 4s^3 + 293917901549477584712119 2s^2 + 342046929585497061253176s + 18557914646800278459054)$ |
| $c_{13}$ | $6(2s + 3)^2(4430976844804689170000 0s^{25} + 9817855551045248780712 00s^{24} + 103571323344221695391126 88s^{23} + 691661317833842749970623 20s^{22} + 32790660951449594262588955 2s^{21} + 11728729364420192968250046 72s^{20} + 32830637216293105459115893 20s^{19} + 73603332392207859667143222 12s^{18} + 13411238091959519801173855 314s^{17} + 2003087433125863944761640 5430s^{16} + 24612632066584435063045693 896s^{15} + 24861807943243247848564702 728s^{14} + 20554616309938676564088193 632s^{13} + 13828701774644445607198489 296s^{12} + 7593307018019776947591316 692s^{11} + 35731644376825396511402989 14s^{10} + 17112054008986347414325887 09s^9 + 10264362013254828732277311 81s^8 + 6933466981581302133514425 87s^7 + 41372946832745234109278382 3s^6 + 1940776438308319265359606 74s^5 + 6856190522023177561905108 0s^4 + 1764097050880333254639794 4s^3 + 313263376945319873280103 2s^2 + 34456042422700093556586 0s + 17745096925489168423620)$ |
| $c_{12}$ | $3(2s + 3)^2(1440067474561523980250 00s^{26} + 33273831804292526086536 00s^{25} + 366868066779219215750244 12s^{24} + 256712663785782687868607 040s^{23} + 12788784884439188697189775 32s^{22} + 48225019656358520783997087 60s^{21} + 14284937167605407625123613 032s^{20} + 3403967573402053121871363 9960s^{19} + 66269194087347804842641890 936s^{18} + 10642025118199905948373959 1464s^{17} + 14167371834788654861152789 6944s^{16} + 15651225373474484983150359 2064s^{15} + 14311945869867279028402035 9156s^{14} + 10777291945795700653556290 3236s^{13} + 66583563316818931905117287 625s^{12} + 34208027977787101480006575 072s^{11} + 15821430510294339891416781 741s^{10} + 80309135220926647430570804 82s^9 + 50505217857341437342571454 60s^8 + 3292374301511579644939102 872s^7 + 18272123652112121151713290 68s^6 + 7951466255776474175898377 40s^5 + 2621421821039038791088123 89s^4 + 6335442721429618093777958 4s^3 + 10625610419721898094320272s^2 + 1108770079900593722922360s + 543716225994186505217 07)$ |

| | |
|---|---|
| $c_{11}$ | $2(2s + 3)^2(2880134949123047960500000s^{27} + 6927926613537598727151600s^{26} + 7968499405072408869622956​0s^{25} + 5830135753486700217327189​84s^{24} + 3044719667796555717818159544s^{23} + 12071152881690852686028406920s^{22} + 37719751802283064561860228312s^{21} + 95186570754684978954315680724s^{20} + 197141443250509137121100298018s^{19} + 338621398294617306936300888486s^{18} + 48532039417125494078362290142s^{17} + 581787906451611153835763035926s^{16} + 582802786083324408675746802192s^{15} + 485993832849417808677870096480s^{14} + 335594444689992516446373541470s^{13} + 191818678234951313275318187166s^{12} + 93215508141687941579846043591s^{11} + 42995488556619072965421127845s^{10} + 22993334205445704498638551659s^{9} + 14702002876383491619167125293s^{8} + 9151802478157350485470780638s^{7} + 4746495171599998113140409294s^{6} + 1930238883493800552821467836s^{5} + 597661705826550277052178582s^{4} + 136369421095118152409287875s^{3} + 21690051407678516824173015s^{2} + 2154400447443907595487183s + 100873131758694046028745)$ |
| $c_{10}$ | $(2s + 3)^2(633629688807070551310000s^{28} + 1584239110567672292139120​0s^{27} + 189762104968446645014104296s^{26} + 1448899233419905865696230​704s^{25} + 7915017749037631812296989272s^{24} + 32911374092963881171128851808s^{23} + 108184113680419836957062571120s^{22} + 288181457405913203915560147656s^{21} + 632563077547125768289724175852s^{20} + 1156966248842167228077853758672s^{19} + 1775610230389152091840398326292s^{18} + 2294661769917892997575576903488s^{17} + 2498192965867701162268688835924s^{16} + 2285659112305489254434518711416s^{15} + 1748991971403770816761602941934s^{14} + 1113797684163785842649673233​76s^{13} + 592491395159743574390096742111s^{12} + 274476600715541565615011213796s^{11} + 127044619585271350650188845452s^{10} + 70538582326147723865407765680s^{9} + 44889681102540017067072498465s^{8} + 26602898887160759567836334520s^{7} + 12967769746561749479001701280s^{6} + 4960789745148078555411299844s^{5} + 145073209873159004959542308​4s^{4} + 3139308944683290821452849​56s^{3} + 475256706206126110586180​19s^{2} + 45067951954440982167499​62s + 2019810664389200880741​21)$ |
| $c_9$ | $2(2s + 3)^2(2880134949123047960500000s^{29} + 7474247118895785746840400s^{28} + 9308513797882609298968418​4s^{27} + 7404022441838906136102386​16s^{26} + 4222492016436341447317422840s^{25} + 18373474587737757143576597976s^{24} + 63374311264983526695777429384s^{23} + 177689296198984770408646605492s^{22} + 411992129847683864453855977890s^{21} + 799270570087512158479167778014s^{20} + 1307454563265970530661662310488s^{19} + 1811440906673507850196226957292s^{18} + 2129008833631551803404401966618s^{17} + 2120318902997465121470552812398s^{16} + 1782752578142774040882126045846s^{15} + 1258348636755929955587480574282s^{14} + 742222950605778500648135120808s^{13} + 368759532677717760323871620448s^{12} + 163292340177616665345297878034s^{11} + 75628135544208165254280371040s^{10} + 428587359546138931024894425​69s^{9} + 26815609860493555439017106967s^{8} + 1515406630759595340156585468​0s^{7} + 698650289823476362744849800​6s^{6} + 2529774938694747662152377363s^{5} + 702330501810674770487862723s^{4} + 144732655027863948827862261s^{3} + 209245472083712624855602​95s^{2} + 1899529777608715114521399s + 8166786538840395351817​5)$ |

| | |
|---|---|
| $c_8$ | $3(2s + 3)^2(14400674745615239802500 0s^{30} + 3873703685787439628342400s^{29} + 5008695060602392586847903 6s^{28} + 41434729334310331141335720 0s^{27} + 2462455716417895553466814404s^{26} + 1119031183906634425953671402 4s^{25} + 40409295972773177050104566556s^{24} + 118947134811306669490272616200s^{23} + 2904604480014499379866096419 84s^{22} + 5956509828769617004442479161 36s^{21} + 10343945033328370448328719736 60s^{20} + 1529122522398341452237748264 328s^{19} + 1929170112144408531295247240352s^{18} + 20772757298357798986279724525 64s^{17} + 1904400092584849147398803257857s^{16} + 1479295019246765244421617960408s^{15} + 967232587581523134128767690737s^{14} + 5298271494562945135798668577 62s^{13} + 245850149371151220247602350838s^{12} + 103586181981415962365079350136s^{11} + 474419698310052107658566514 90s^{10} + 27017370328613731242988726116s^9 + 165436040746213423165348338 96s^8 + 89598081048155039681209011 60s^7 + 393414848050341758787717765 3s^6 + 13567085019266032207440806 94s^5 + 3593560100508123307874732 79s^4 + 7080039802536542890191925 2s^3 + 980570563944186001099307 4s^2 + 8542944279890061972050 52s + 3530618607561154024305 6)$ |
| $c_7$ | $12(2s + 3)^2(22154884224023445850000s^{31} + 616966740327228674348400s^{30} + 8270907073696162683430488s^{29} + 7105488837495844741933140 0s^{28} + 43931915711054680381189696 8s^{27} + 20811530357045666581306248 00s^{26} + 78516994084586808022426344 84s^{25} + 242075167606254230722070734 14s^{24} + 62092517428805875866794674881s^{23} + 134191349159021239688414857431s^{22} + 246518378732311775029464265197s^{21} + 387229048800653531738114739999s^{20} + 521849077561894657636398449655s^{19} + 604010499659026951460897607741s^{18} + 599706193952136964321127694249s^{17} + 508932600092700362308014694866s^{16} + 366923845830106735087199088303s^{15} + 222970678878271885535159028828s^{14} + 113510751404816691589761355815s^{13} + 489596215801597338845289884633s^{12} + 193438197331709555443195521 63s^{11} + 857096764824341929866037721 1s^{10} + 48422817052135271084165132 79s^9 + 290913837456231799036003139 4s^8 + 152292130359617399613857590 5s^7 + 642212965528640572310436879s^6 + 21234740879260983664481535 9s^5 + 53940504869138165983355556s^4 + 102009814215488662722720 21s^3 + 13576094237482264271577 78s^2 + 11378258779245747678886 5s + 4528476210896135134182)$ |
| $c_6$ | $3(2s + 3)^2(44309768448046891700000s^{32} + 127595813491277942713440 0s^{31} + 17712124648743520374246000s^{30} + 15780078330319234248480777 6s^{29} + 10134760020968226539342475 36s^{28} + 4996346811471994438240970400s^{27} + 19656891049003759997738219664s^{26} + 63343018585986823082619729288s^{25} + 170258537043632779229473723584s^{24} + 386719255970188993917224574336s^{23} + 749195973037122202127202135768s^{22} + 1245952789865163842159925500256s^{21} + 178599096936071504600001173635 68s^{20} + 2210911897085152430737822315488s^{19} + 2363325656949833631869930474232s^{18} + 2176386534475868543250178452024s^{17} + 1718519269597730769046219931925s^{16} + 1154898997656814617038309424924s^{15} + 653993799682370716585040976045s^{14} + 309097812387589842994049088882s^{13} + 122649986514805927975811013678s^{12} + 44139321593386486437541304232s^{11} + 18194600512073131179652474218s^{10} + 100807294552471771494663601 08s^9 + 60121544950098328631438607 03s^8 + 30867139019044853020774239 12s^7 + 12645365130026448915535735 32s^6 + 40426900431353599770906418 8s^5 + 990853185541373958934672 65s^4 + 18066869592965845573731768s^3 + 23180375592196844545338 93s^2 + 1873358777849595772989 78s + 719206981536479606622 9)$ |

| | |
|---|---|
| $c_5$ | $6(2s + 3)^2(8861953689609378340000s^{33} + 2635965557834220301114400s^{32} + 3784460184258343211722032s^{31} + 34920466929143934879808368s^{30} + 232642130166877120425752976s^{29} + 1191697800945367086458055072s^{28} + 4880694751880441559218476632s^{27} + 16406950824469547680560778764s^{26} + 46112865280418851489322074134s^{25} + 109812274944960351155991786018s^{24} + 223725856433516713998849805632s^{23} + 392660140917098227883941019880s^{22} + 596450332899276085006754000436s^{21} + 786241559139425306167747437324s^{20} + 900162472445706270827205735408s^{19} + 894118456198813762374744874842s^{18} + 768019765700356821541048064211s^{17} + 567188919515029404795124639815s^{16} + 356892849173690588784279865923s^{15} + 188887726440058090918123569807s^{14} + 82758519677573937797801658678s^{13} + 29755841968295155180938377160s^{12} + 9265655123985315388229427762s^{11} + 3269804705124270960946179492s^{10} + 1745378750437674672254525343s^9 + 1067124171571276165079329161s^8 + 553192327447935426146669544s^7 + 224471881547232310097558892s^6 + 70271287113829333092849234s^5 + 1675985651729512089436656s^4 + 2963427571961631174417201s^3 + 367988093962382467875321s^2 + 28750493973686584294998s + 1066338997876680421860)$ |
| $c_4$ | $3(2s + 3)^2(5538721056005861462500s^{34} + 170000930428677948001200s^{33} + 2521542870629614052494182s^{32} + 24069033861073900393194288s^{31} + 166112140239471730224127950s^{30} + 882861984743248195220227068s^{29} + 3758135559481861075860190560s^{28} + 13155787658258741235890755800s^{27} + 38587134973448786366330248032s^{26} + 96129109345583765886638321232s^{25} + 205447147746380335664768754954s^{24} + 379449453375926778871028692368s^{23} + 608776780980435381145177931646s^{22} + 851251839731897497067600541072s^{21} + 1039128387112340409627618422211s^{20} + 1107338907683241287910817574616s^{19} + 1028236425910911889110211542327s^{18} + 828653388719550997841334599814s^{17} + 575679860099214460991336767122s^{16} + 341113980250874385715195498752s^{15} + 169602112160966873244672554562s^{14} + 69019121880524178108395152092s^{13} + 22192275965509681657763172138s^{12} + 5513039324469607430622101184s^{11} + 1308552733582056616685717799s^{10} + 622822426968272935610750562s^9 + 441208218615564366021478680s^8 + 251086626744997918806221832s^7 + 105542726343197962447668330s^6 + 33182712041751038192684904s^5 + 7819078036046387444720541s^4 + 1353372939412637489000388s^3 + 163593372233172281873202s^2 + 12396727299661082512092s + 444813790293506728368)$ |
| $c_3$ | $6(2s + 3)^2(651614241883042525000s^{35} + 20618119083643318565400s^{34} + 315621334999692786151116s^{33} + 3113065178572047666007860s^{32} + 22229797545183952680520284s^{31} + 122422561739941481599835484s^{30} + 540840514528742111761776912s^{29} + 1968384017248063183323104976s^{28} + 6014334916396782599305697730s^{27} + 15642733509059107994947830402s^{26} + 34991063427920339849664070098s^{25} + 67834512242456744868412053510s^{24} + 114610735641635823232680558264s^{23} + 169420289219412443682245006436s^{22} + 219630129457905943676457475806s^{21} + 249914888458917575863280485878s^{20} + 249458032509437740525240777731s^{19} + 217920380868106493481515756421s^{18} + 165868299839291545410161133771s^{17} + 109196804131621756456248426789s^{16} + 61455743971044438451107683232s^{15} + 29010820857450160933812140136s^{14} + 111111724769820435809112639700s^{13} + 3230978841571830756780993930s^{12} + 597734071316630030096071452s^{11} + 19471456710454363005152664s^{10} - 14413823109350224541499726s^9 + 9684588377731643071927236s^8 + 12560219583039185504039910s^7 + 6629813221106696409536742s^6 + 2267528918316638233964400s^5 + 549510851612447197946100s^4 + 95169593716835317546698s^3 + 11333379076369208961606s^2 + 837761545501050427146s + 29118359247617829474)$ |

| | |
|---|---|
| $c_2$ | $(2s + 3)^2(65161424188304252500 0s^{36} + 212361287050892314836 00s^{35} + 3351768289135045172584 92s^{34} + 34124802723490603145907 60s^{33} + 25184187621847674304218 612s^{32} + 143532477912633204107111 040s^{31} + 6571995334134877377460058 40s^{30} + 248305780276214198137260 4872s^{29} + 7890444256414221204984097 182s^{28} + 2138670662626443899749925 7504s^{27} + 4996862660986503205494353 2722s^{26} + 1014439442075341249520382 83748s^{25} + 1800227084981344254247742 57358s^{24} + 2804733165486533307037649 39232s^{23} + 3847701887175612902111068 82724s^{22} + 4655658061409025346140795 055268s^{21} + 4970705288178906475764968 74221s^{20} + 4678534555507223380165602 91068s^{19} + 3872550916392453613427525 51670s^{18} + 2806584923777792155024177 61676s^{17} + 1768405807430274222159766 20477s^{16} + 9579106278655550872408874 2320s^{15} + 4378586233016249905220952 9768s^{14} + 1632673685352712299171515 5824s^{13} + 4608219050084326790748933 153s^{12} + 7630908802856105790078631 08s^{11} - 7027359523643236832970771 6s^{10} - 9985407270432287153739260 4s^9 - 357260313779697920881889 25s^8 - 609903789530767014228760 8s^7 + 38599700071122383235616 8s^6 + 57180955054180793753095 2s^5 + 18478934378953446115053 0s^4 + 3548837556962247216924 0s^3 + 436295085881317044922 8s^2 + 320602140994390122456s + 10806813741383936712)$ |
| $c_1$ | $2s^2(2s + 3)^2(5s + 6)^2(137181945659587900 0s^{33} + 4271634557055966808 8s^{32} + 643544475178313701908 s^{31} + 6247566555702580389060 s^{30} + 4391676556526924901666 0s^{29} + 2381294370524934521705 40s^{28} + 1036074226442125183419 168s^{27} + 3714930106381256786671 824s^{26} + 1118772868707257505376 3704s^{25} + 2869670825417555723588 6484s^{24} + 6335289140657084244295 6878s^{23} + 1213301616269188486575 58398s^{22} + 2027652551311758119054 86752s^{21} + 2969521561596848429994 76188s^{20} + 3821939060079245330600 66196s^{19} + 4329771063731500080610 33636s^{18} + 4318829425555373349451 82376s^{17} + 3789227783217399510763 19160s^{16} + 2916937858513690287806 62644s^{15} + 1961472241580118928394 33394s^{14} + 1144103496246446264232 12729s^{13} + 5724888433243097639645 1627s^{12} + 2413061795050375698405 1008s^{11} + 8287189086050003227856 022s^{10} + 2152391916458370195915 867s^9 + 325184614597411140314 601s^8 - 330079723518789034754 04s^7 - 417925109386866638897 304s^6 - 15831185972996449730 358s^5 - 3877006197061115690 130s^4 - 665506024092175855 680s^3 - 78419814392209911120 s^2 - 5759186746951521828s - 200126180395998828)$ |
| $c_0$ | $s^4(2s+3)^2(5s+6)^4(5487277826383516 s^{30} + 162900207047936448 s^{29} + 2337377142714373098 s^{28} + 21588165729897598296 s^{27} + 144210704373637237422 s^{26} + 742206852421449807276 s^{25} + 3061285798160471289822 s^{24} + 10391895636644586020112 s^{23} + 29588363727735612069036 s^{22} + 71651404108146138897096 s^{21} + 149117620073027461547436 s^{20} + 268806705178958727187248 s^{19} + 422185992215286625454736 s^{18} + 580191752386498986323712 s^{17} + 699694724310231624064272 s^{16} + 741757261801656111225984 s^{15} + 691666247103583662351612 s^{14} + 567033087779716710023352 s^{13} + 408047131644656580559296 s^{12} + 257036644794565634383008 s^{11} + 141145264141826804073576 s^{10} + 67178460532288884909516 s^9 + 27499663057942951141041 s^8 + 9582491278489242855672 s^7 + 2803365139419823150782 s^6 + 675682953313836552876 s^5 + 130659498671205647052 s^4 + 19489563056909654496 s^3 + 2105305456045150908 s^2 + 146594486051390088s + 4941387170271576)$ |

$$\Delta_2 \quad \begin{aligned}
&-36s^4(2s+3)^4(5s+6)^4(18818886214950127475987197826410000s^{66}\\
&+1171982279367338935485204891772187040000s^{65}\\
&+359034059515843764915172404946627408992s^{64}\\
&+7212088571266452516322419090023965301952s^{63}\\
&+106839132025368784093448214456114080089896s^{62}\\
&+1244653930745343462400612341123366430556112s^{61}\\
&+11874531272586759157048871425273776419646480s^{60}\\
&+95397174639780849117400217925290860797543072s^{59}\\
&+658601421196162394675006092409270655783387096s^{58}\\
&+3968022355733696091765263130542489256493297296s^{57}\\
&+21117182157874536980169655465156091744276920380s^{56}\\
&+100234695532226306436649215523120163350432040784s^{55}\\
&+427723545408455739277440670239201960498527445168s^{54}\\
&+1651693204898583110001983831675383703838803111280s^{53}\\
&+5803881305128515928064472759889503906134840719788s^{52}\\
&+18645262399111732404927179918528703191829668776944s^{51}\\
&+54982160923970862907155853987157692828051002129468s^{50}\\
&+149340327131211909506021191619303279476069307001480s^{49}\\
&+374738908504925756556865067810806785110272348870998s^{48}\\
&+870959171363388767125455183938474548684433032733976s^{47}\\
&+1879125423105677523532695113591118653251030543219650s^{46}\\
&+3770893425502604495219029177077658520798571516698604s^{45}\\
&+7050015221784174490794108388251813073259643663091008s^{44}\\
&+12297516978145400887377430125405972391309535236404816s^{43}\\
&+20038164484164236659289158464333223322979861456369558s^{42}\\
&+30532204930658301922179960307897986850960973110401060s^{41}\\
&+43539740165642307747449376716152274738997971952381288s^{40}\\
&+58147942722340474688455284463585466926829447124690864s^{39}\\
&+72765609819670245981642226229421333542816165300786478s^{38}\\
&+85352226271944117403522054472274990615987412077658020s^{37}\\
&+93861185123580180986528832241395054124125210332787978s^{36}\\
&+96773676228707454794185949836280770022200702637789136s^{35}\\
&+93535505340566829835280969592461902479874869737070324s^{34}\\
&+84726573070331222224010481384087646174794999517275720s^{33}\\
&+71892923025396467242385567135204296459367606931171072s^{32}\\
&+57107505385821670765861094421169358296399126222638720s^{31}\\
&+42429475579369200294980396815322839839865021711481712s^{30}\\
&+29453402248671311370918975586371950400100875859184424s^{29}\\
&+19076887418815475591391225344828673331985433949859745s^{28}\\
&+11509630902613742430056710113025979946368929363591560s^{27}\\
&+6455314570589262935425947995355398219841083619460275s^{26}\\
&+3357432561061020603580382785922585900536396393910862s^{25}\\
&+1614483581569691108434974127018929362203153022365980s^{24}\\
&+715184392123946764432274154396140356129136774278152s^{23}\\
&+290561593707882411154971270303899134550457836920080s^{22}\\
&+107686614725038641896140906093508540897806554334884s^{21}\\
&+36175123767827876571287168571772096066641842189715s^{20}\\
&+10936295413621417840147244658584579127207174389352s^{19}\\
&+2956231446951945885452453177600650222360031507853s^{18}\\
&+714328315495087334767452511829810830341704532294s^{17}\\
&+157862256397341201315796262528952028945210160178s^{16}\\
&+34669184815408932243484290688498684622916000552s^{15}\\
&+8792208859688249070338942374434094956929968434s^{14}\\
&+2728958913497904679605629304348283067662149156s^{13}\\
&+908514477310679518991239534084776498569549130s^{12}\\
&+281997384651571824431416925444015838101696600s^{11}\\
&+765957125613299444411389424541888727384 96992s^{10}
\end{aligned}$$

$$+17823235368805916505433544016195415996681584s^9$$
$$+35225074079403593877757698013822148284205165s^8$$
$$+586782005064608000832043474107919395519552s^7$$
$$+81484523152763229188477555846803640661672s^6$$
$$+9275689993243887053899463828498929963344s^5$$
$$+843808167321923057379954129881612634096s^4$$
$$+58988087589412129623633466476137297856s^3$$
$$+29725812874330715681181158507796330725^2$$
$$+959233912036916287714016721581540165$$
$$+148335141036636539337219080656939^2)$$

The last lemma presents some properties for the population-level SGPO and GRPO dynamics.

**Lemma B.4.** *Under the assumptions from Theorem 3.3, the following statements hold true,*

*(i)* $p_{SGPO}^{(k)}, q_{SGPO}^{(k)}, p_{GRPO}^{(k)}, q_{GRPO}^{(k)} \in (0,1)$ *for all* $k \geq 0$.
*(ii)* $p_{SGPO}^{(k)}, q_{SGPO}^{(k)}, p_{GRPO}^{(k)}, q_{GRPO}^{(k)}$ *are strictly increasing in* $k$ *and lie in* $(\frac{1}{2}, 1)$ *for all* $k \geq 1$.
*(iii)* $p_{SGPO}^{(k)} > q_{SGPO}^{(k)}$ *for all* $k \geq 1$.

*Proof.* We first rewrite the update rule in Eq. (3) as follows,

$$p_{\text{SGPO}}^{(k+1)} = p_{\text{SGPO}}^{(k)} \frac{e^{\Delta_{\text{SGPO},p}^{(k)}}}{1 - p_{\text{SGPO}}^{(k)} + p_{\text{SGPO}}^{(k)} e^{\Delta_{\text{SGPO},p}^{(k)}}}, \quad \text{where} \quad \Delta_{\text{SGPO},p}^{(k)} = p_{\text{SGPO}}^{(k)}(1 - p_{\text{SGPO}}^{(k)}),$$

$$q_{\text{SGPO}}^{(k+1)} = q_{\text{SGPO}}^{(k)} \frac{e^{\Delta_{\text{SGPO},q}^{(k)}}}{1 - q_{\text{SGPO}}^{(k)} + q_{\text{SGPO}}^{(k)} e^{\Delta_{\text{SGPO},q}^{(k)}}}, \quad \text{where} \quad \Delta_{\text{SGPO},q}^{(k)} = (p_{\text{SGPO}}^{(k)})^2 q_{\text{SGPO}}^{(k)}(1 - q_{\text{SGPO}}^{(k)}),$$

$$p_{\text{GRPO}}^{(k+1)} = p_{\text{GRPO}}^{(k)} \frac{e^{\Delta_{\text{GRPO},p}^{(k)}}}{1 - p_{\text{GRPO}}^{(k)} + p_{\text{GRPO}}^{(k)} e^{\Delta_{\text{GRPO},p}^{(k)}}}, \quad \text{where} \quad \Delta_{\text{GRPO},p}^{(k)} = p_{\text{GRPO}}^{(k)}(1 - p_{\text{GRPO}}^{(k)}) q_{\text{GRPO}}^{(k)},$$

$$q_{\text{GRPO}}^{(k+1)} = q_{\text{GRPO}}^{(k)} \frac{e^{\Delta_{\text{GRPO},q}^{(k)}}}{1 - q_{\text{GRPO}}^{(k)} + q_{\text{GRPO}}^{(k)} e^{\Delta_{\text{GRPO},q}^{(k)}}}, \quad \text{where} \quad \Delta_{\text{GRPO},q}^{(k)} = p_{\text{GRPO}}^{(k)} q_{\text{GRPO}}^{(k)}(1 - q_{\text{GRPO}}^{(k)}).$$

First of all, the uniform initialization yields the desired result for $k = 0$. Suppose $p_{\text{SGPO}}^{(k)} \in (0,1)$ for some $k \geq 0$. Then, we have

$$1 - p_{\text{SGPO}}^{(k)} + p_{\text{SGPO}}^{(k)} e^{\Delta_{\text{SGPO},p}^{(k)}} > p_{\text{SGPO}}^{(k)} e^{\Delta_{\text{SGPO},p}^{(k)}} > 0,$$

which implies $p_{\text{SGPO}}^{(k+1)} \in (0,1)$. By induction, we have $p_{\text{SGPO}}^{(k)} \in (0,1)$ for all $k \geq 0$. Similarly, we can show that $q_{\text{SGPO}}^{(k)}, p_{\text{GRPO}}^{(k)}, q_{\text{GRPO}}^{(k)} \in (0,1)$ for all $k \geq 0$.

Furthermore, we have $\Delta_{\text{SGPO},p}^{(k)} > 0$ since $p_{\text{SGPO}}^{(k)} \in (0,1)$. This implies

$$\frac{p_{\text{SGPO}}^{(k+1)}}{p_{\text{SGPO}}^{(k)}} = \frac{1}{(1 - p_{\text{SGPO}}^{(k)}) e^{-\Delta_{\text{SGPO},p}^{(k)}} + p_{\text{SGPO}}^{(k)}} > \frac{1}{1 - p_{\text{SGPO}}^{(k)} + p_{\text{SGPO}}^{(k)}} = 1.$$

Since $p_{\text{SGPO}}^{(0)} = \frac{1}{2}$, we have $p_{\text{SGPO}}^{(k)} \in (\frac{1}{2}, 1)$ for all $k \geq 1$. Similarly, we can show that $q_{\text{SGPO}}^{(k)}, p_{\text{GRPO}}^{(k)}, q_{\text{GRPO}}^{(k)}$ are strictly increasing and lie in $(\frac{1}{2}, 1)$.

Finally, we have $p_{\text{SGPO}}^{(0)} \geq q_{\text{SGPO}}^{(0)}$. Thus, it suffices to show that $p_{\text{SGPO}}^{(k)} \geq q_{\text{SGPO}}^{(k)}$ implies $p_{\text{SGPO}}^{(k+1)} > q_{\text{SGPO}}^{(k+1)}$ for all $k \geq 0$. Indeed, Lemma B.1(i) and $p_{\text{SGPO}}^{(k)} \geq q_{\text{SGPO}}^{(k)}$ yield

$$p_{\text{SGPO}}^{(k+1)} = \exp(f_{11}(p_{\text{SGPO}}^{(k)})) \geq \exp(f_{11}(q_{\text{SGPO}}^{(k)})) = \exp(\log(q_{\text{SGPO}}^{(k)}) + h_{q_{\text{SGPO}}^{(k)}}(q_{\text{SGPO}}^{(k)}(1 - q_{\text{SGPO}}^{(k)}))).$$

Then, Lemma B.1(ii) and $p_{\text{SGPO}}^{(k)} \in (0,1)$ yield

$$\exp(\log(q_{\text{SGPO}}^{(k)}) + h_{q_{\text{SGPO}}^{(k)}}(q_{\text{SGPO}}^{(k)}(1 - q_{\text{SGPO}}^{(k)}))) > \exp(\log q_{\text{SGPO}}^{(k)} + h_{q_{\text{SGPO}}^{(k)}}((p_{\text{SGPO}}^{(k)})^2 q_{\text{SGPO}}^{(k)}(1 - q_{\text{SGPO}}^{(k)}))).$$

In addition, we have

$$q_{\mathtt{SGPO}}^{(k+1)} = \exp(f_{12}(p_{\mathtt{SGPO}}^{(k)}, q_{\mathtt{SGPO}}^{(k)})) = \exp(\log q_{\mathtt{SGPO}}^{(k)} + h_{q_{\mathtt{SGPO}}^{(k)}}((p_{\mathtt{SGPO}}^{(k)})^2 q_{\mathtt{SGPO}}^{(k)}(1 - q_{\mathtt{SGPO}}^{(k)}))).$$

Putting these pieces together yields $p_{\mathtt{SGPO}}^{(k+1)} > q_{\mathtt{SGPO}}^{(k+1)}$. $\qquad\square$

