# OpenReview forum: "Stepwise Guided Policy Optimization: Coloring Your Incorrect Reasoning in GRPO"
_TMLR — Accepted by TMLR_

### Review · Reviewer_UtfL · 2025-12-16

**Summary Of Contributions:**

This paper highlights the issue with certain modern GRPO methods, that gradients are zero when all reasoning trajectories in a group represent failures, and then proposes an alternative method that solves this issue. Specifically, the authors propose a “step-wise” approach to group-relative policy optimization, where the rewards for performance are not given solely based on the ultimate success or failure of a reasoning trajectory, but based on the number of correct intermediate reasoning steps taken by the model. To obtain an estimate of the correct number of reasoning steps, the authors propose using a second “judge” model to estimate the correct steps in each trajectory. The authors evaluate their method numerically, primarily by comparing with GRPO, and theoretically, by comparing their method and GRPO via a toy model. They show that their model tends to perform better than GRPO, by using information gained from partial correctness on a reasoning task.

## Strengths
I believe there are two primary strengths of the paper. First is its effort to include theoretical analysis. Including theory, even with a toy model, can provide complementary insights that one might miss with purely numerical work, so I appreciate the authors’ efforts on this front. Second, I think the solution that the authors propose is elegant in its simplicity.

## Weaknesses
In my view, the primary weaknesses of the paper are (1) not benchmarking relative to existing process models; (2) not always quantifying statistical significance (or variance of measurements) in empirical results, and limited discussion of variability in the results; (3) some issues with the theoretical results, particularly a lack of derivations for some of the main equations in the lead up to the main theorem and a toy model that seems to miss some of the more important aspects of the algorithm being studied; (4) clarity of presentation. I will elaborate on these issues in the requested changes section.

**Additional Comments:**

I have two meta-notes regarding my review. First, reasoning in large models is not my domain of expertise, so please take this into account when considering my review. Second, I have not verified the proof of the main theorem. The reason for this is that I have some immediate problems with other aspects of the paper (see main weaknesses) that I believe need to be resolved. If the authors are able to sufficiently address these other issues I will then be happy to check through the proof.

**Audience:**

Yes

**Audience Explanation:**

Given modern interest in reasoning, LLMs, and feedback during learning, the subject matter of this paper is definitely of interest to many in the community. For this reason, I have answered "yes" above. However, because of issues around baseline choices (mentioned below) I have some concerns that researchers may be less interested in the papers results despite the subject matter.

**Broader Impact Concerns:**

In addition to potential societal benefits of large AI models, there is a growing body of literature observing certain, seemingly severe, negative impacts—including energy consumption, power/wealth concentration, job loss, and threats to democracy. For easily digestible overviews of such impacts, one could look at the Atlas of AI by Kate Crawford or The AI Con by Hanna and Bender. For academic articles one could start with works by the same authors. Given this growing literature, and given the focus of the article on improving large reasoning models, I was surprised to see no mention of broader impact concerns in the paper. I would like to see at least some brief discussion of broader impacts in the paper before I can recommend it for publication.

**Claims And Evidence:**

No

**Claims Explanation:**

I do not believe the “consistent gains” claim made in the paper is sufficiently supported. This is because of variability in the results, and baseline choices.

**Requested Changes:**

I list the requested changes below. I have added “**C**” at the end of changes that I view as critical, for my review, and “**S**” after ones that I view more as suggestions. I have also divided my review up into sections based on each of the three weaknesses mentioned in ‘strengths and weaknesses’, above, and a fourth section highlighting minor (mostly spelling and grammar) issues.

**Weakness 1: benchmarking and comparison to other models**
1. The big issue the paper aims to address is the lack of partial rewards for groups with no successful reasoning trajectories that one sees in SGPO. As the authors mention, there are already existing methods that aim to resolve this issue—e.g. process reward models. While, as discussed in remark 3.1, there seem to be differences between their approach and these other methods, it seems to me that the correct benchmark in the paper is not GRPO but, rather, these other methods that already aim to address the problem of lack of partial rewards. Additionally, GRPO is known to exhibit certain problems (e.g. bias towards long response lengths), as mentioned in “Understanding R1-Zero-Like Training: A Critical Perspective” by Liu et al. For these reasons, I would like to see a comparison between SGPO and existing process reward type methods—instead of GRPO—either demonstrating the utility of SGPO from a performance or computational efficiency perspective. For example, “VinePPO: Refining Credit Assignment in RL Training of LLMs”, by Kazemnejad et al. could be worth looking at as a benchmark. **C**
2. [Remark 3.1, (i)]: it seems that the step-wise value that the paper studies could just be represented as a simple value function that assigns monotonically increasing value with each correct step, no? **C**

**Weakness 2: statistical significance, and variability in results**
1. Many of the results presented in the paper do not have measures of statistical significance—for example tables 1 and 2, and figures 2, and 3. Given the, at times, small differences in performance between models, it would be really nice to have some quantification of whether or not result differences are significant. Even a measure of standard deviation could help with this. **C**
2. It does not seem that the authors’ model, SGPO, universally outperforms GRPO—or even the baseline—in terms of performance (see e.g. Table 2). One would have expected, at first glance, that having the extra information from all-negative samples should, at worst, not drop below the baselines that do not address the negative sample issue. However, there is not much discussion of this in the paper. It would be good to have some deeper discussion of when and why the authors’ approach does not outperform these alternatives.  **C**
3. It is claimed that $\beta$ and $\gamma$ filter and smooth noisy signals and that removing them weakens performance. Three questions related to this: first, are you using filtering and smoothing in the usual statistical time-series sense? If so, how are they performing these operations? Second, it is not clear to me that removing $\beta$ and $\gama$ reduces performance or increases variance in a statistically significant fashion. It would be good to have a statistical test to confirm this. Third, when you say you “remove” these parameters, you mean setting them equal to $1$ and $0$, correct? **C**

**Weakness 3: theoretical results**
1. Importance sampling and clipping is not included in the model. This is clearly done to make analysis more simple, but, as far as I have seen, there is no description of the impact of these simplifications. It would be great to at least speculate about this a little bit. **S**
2. The theoretical section focuses on proving that including information in learning for partially correct answers yields benefits over not including it. However, this seems somewhat intuitive. To me, the bigger theoretical question associated with the authors’ new method is how using other large models as judges—even when they may provide potentially incorrect insight on partial correctness—can lead to failures or successes in learning. Have the authors considered ways of investigating this question? **S**
3. In the third line of equations on page 6 the authors get factors of $½$ multiplying each gradient; I have tried to rederive these and obtain different factors (namely $2$ for GRPO and $2$ and $4$ respectively for the top two and bottom two elements of the gradient for SGPO). Could the authors add a section deriving these gradient forms to the appendix please? **C**
4. Would it be possible to add derivations for equations (3) and (4) in the appendix of the paper? **C**

**Weakness 4: clarity of presentation**
1. Regarding reference solutions: to clarify, must there be a reference solution for every question in the dataset on which the model is trained? I.e. SGPO requires training datasets with fully answered train of thought solutions? **C**
2. [section 4.1]: What do “avg@16” and “pass@16” mean? It could be good to briefly describe these. **C**
3. [appendix A]: I wonder if the “related works” section could be more logically organized—it seems like the text jumps back and forth between concepts a little bit. For example, chain of thought reasoning is referenced before the “chain of thought and its variants” section, which makes me wonder if the chain of thought section should come earlier. **C**
4. Perhaps it is just me, but I am somewhat confused by the choice of “colouring” in the title. What is the rationale for this? **S**


**Minor issues**
1. [Intro line 8 and elsewhere]: There are many places in the paper where the authors add “the” where it is not needed. For example, line 8 of introduction: “As *the* generative AI applications…”
2. [Intro line 9]: “conversational interfaces” is a very strange phrasing; suggest reword.
3. [Intro line 9]: “increasingly powerful” => more powerful
4. [Intro line 10]: what does “...positioning them as a key frontier in practice” mean? Suggest rephrasing this
5. [end of page 1]: first mention of “group-relative” training. I suggest including a quick description of this
6. [page 2, top of 3rd paragraph]: “The key insight is that many reasoning tasks possess a structure where step-level correctness can be explicitly defined.” The key insight where? In the above paragraph or in your own work?
7. [first line, page 3]: distinguish => distinguished
8. [section 2]: could be useful to start by mentioning the focus on fine-tuning.
9. [3rd equation in section 2]: define “clip” operation
10. [Equation 2]: Why use the sigmoid function? If RTS is zero, shouldn’t the reward be zero?
11. [Example in Remark 2]: the example starting with “The twelve letters {A, B, C, D, E, F, G, H, I, J, K, L} are randomly grouped” seems redundant.
12. [2nd line after first equation section 3.2]: $\theta$ in the definition of the score should be a subscript, right?
13. In some of the results tables it could be useful to present results as bar plots. The tables are quite dense and difficult to take in at times.
14. [Table 5]: what does “SGPO\GRPO”, and vice versa, mean in the table heading? Somewhat confusing.
15. [page 23]: what do you mean by “given that the transition is deterministic”? Are deterministic transitions required for DPO?
16. [page 23]: when you say “the specific loss using a prospect theory…”, what do you mean by “specific loss”? Do you mean the DPO loss?

---

> ### Author Response · Authors · 2026-01-11
> **Author response to reviewer (part 1/4)**
>
> Thank you for the careful reading and the concrete suggestions. We try to response point by point as follows.
>
> 1. **(Benchmarking and Comparison to Other Model) It seems to me that the correct benchmark in the paper is not GRPO but, rather, other methods that already aim to address the problem of lack of partial rewards. I would like to see a comparison between SGPO and existing process reward type methods &mdash; instead of GRPO &mdash; either demonstrating the utility of SGPO from a performance or computational efficiency perspective. For example, "*VinePPO: Refining Credit Assignment in RL Training of LLMs*", by Kazemnejad et al. could be worth looking at as a benchmark.**
>
> We agree that process-reward approaches are also relevant. SGPO, however, is specifically designed as an improvement to GRPO, modifying only the within-group advantage construction while leaving the rest of the training pipeline unchanged. For this reason, GRPO is the most direct and controlled baseline to isolate our contribution. We have cited VinePPO (Kazemnejad et al.) and added a discussion of its relationship to SGPO. Conceptually, VinePPO provides intermediate rewards via Monte Carlo value estimation in a PPO-style framework, whereas SGPO introduces post-hoc first-error localization to construct a single scalar reward within GRPO. A full empirical comparison would require substantial additional experimentation and is left to future work.
>
> 2. **(Benchmarking and Comparison to Other Model) It seems that the step-wise value that the paper studies could just be represented as a simple value function that assigns monotonically increasing value with each correct step, no?**
>
> This is an excellent suggestion. One can indeed view the step-wise score as a monotone quantity increasing with the number of verified correct steps. We do not call it a value function since, unlike PRMs, the score is not used to guide search or decoding, nor to approximate expected future returns. Instead, it is used only to construct a single scalar reward for policy updates after the full trajectory is observed.
>
> 3. **(Statistical Significance and Variability) Many of the results presented in the paper do not have measures of statistical significance &mdash; for example Tables 1 and 2, and Figures 2, and 3. Given the, at times, small differences in performance between models, it would be really nice to have some quantification of whether or not result differences are significant. Even a measure of standard deviation could help with this.**
>
> We agree that reporting variability strengthens the evaluation. Due to computational constraints, we could not rerun all experiments multiple times. That being said, we report mean and standard deviation over multiple runs for a subset of settings (Tables 3 and 4), and we acknowledge more comprehensive significance analysis as future work.
>
> 4. **(Statistical Significance and Variability) It does not seem that the authors' model, SGPO, universally outperforms GRPO &mdash; or even the baseline &mdash; in terms of performance (see e.g.  Table 2). One would have expected, at first glance, that having the extra information from all-negative samples should, at worst, not drop below the baselines that do not address the negative sample issue. However, there is not much discussion of this in the paper. It would be good to have some deeper discussion of when and why the authors' approach does not outperform these alternatives.**
>
> We have added discussion clarifying that SGPO does not guarantee uniform improvements. In some cases, partial-credit signals may introduce noise or be less informative, particularly when negative samples lack meaningful structure. SGPO is most beneficial when failures are near-misses rather than fundamentally flawed reasoning. This dependence on failure structure is now discussed explicitly.

---

> ### Author Response · Authors · 2026-01-11
> **Author response to reviewer (part 2/4)**
>
> 5. **(Statistical Significance and Variability) It is claimed that $\beta$ and $\gamma$ filter and smooth noisy signals and that removing them weakens performance. Three questions related to this: first, are you using filtering and smoothing in the usual statistical time-series sense? If so, how are they performing these operations? Second, it is not clear to me that removing $\beta$ and $\gamma$ reduces performance or increases variance in a statistically significant fashion. It would be good to have a statistical test to confirm this. Third, when you say you "remove" these parameters, you mean setting them equal to $1$ and $0$, correct?**
>
> Our "smoothing" is not time-series filtering. It is a static calibration applied per trajectory. Specifically, we compute a reasoning-trajectory score $\texttt{RTS}(\mathbf{y}) \in [0, 1]$, defined as the fraction of steps verified correct up to the first error, and map it to a scalar reward via a sigmoid with $\beta=10$ and $\gamma=0.5$. This mapping sharpens the distinction between early failures and near-miss trajectories, reducing sensitivity to judge noise around the first-error boundary.
>
> We agree that this is a heuristic rather than a statistically guaranteed procedure. That said, we report mean and std over multiple runs for representative settings (Table 3, Figure 2), including an ablation without $(\beta, \gamma)$, which shows higher variance and slightly weaker performance. When we say we "remove" these parameters, we remove the sigmoid entirely and use the raw RTS score $\texttt{RTS}(\mathbf{y})$ as the reward for incorrect samples.
>
> 6. **(Theory) Importance sampling and clipping is not included in the model. This is clearly done to make analysis more simple, but, as far as I have seen, there is no description of the impact of these simplifications. It would be great to at least speculate about this a little bit.**
>
> We omit importance sampling and clipping to keep the analysis tractable and isolate the effect of reward diversification in all-negative groups. In practice, both mechanisms are important for stability; we note this explicitly and leave their principled theoretical treatment to future work. Qualitatively, they affect update magnitude and variance, whereas our theoretical separation hinges on whether the group-normalized advantage vanishes under different reward designs.
>
> 7. **(Theory) The theoretical section focuses on proving that including information in learning for partially correct answers yields benefits over not including it. However, this seems somewhat intuitive. To me, the bigger theoretical question associated with the authors' new method is how using other large models as judges  &mdash; even when they may provide potentially incorrect insight on partial correctness &mdash; can lead to failures or successes in learning. Have the authors considered ways of investigating this question?**
>
> We agree this is the central theoretical question. In this paper, we intentionally analyze an idealized setting with a reliable step-wise judge to isolate the effect of partial-correctness signals. Even in this regime, obtaining a clean per-iteration separation between SGPO and GRPO requires a stylized setting and does not yet extend to general horizons or action spaces. Analyzing noisy or biased judges would require modeling judge errors and their interaction with group-normalized updates, which is substantially more challenging and left to future work.
>
> 8. **(Theory) In the third line of equations on page 6 the authors get factors of 1/2 multiplying each gradient; I have tried to rederive these and obtain different factors (namely 2 for GRPO and 2 and 4 respectively for the top two and bottom two elements of the gradient for SGPO). Could the authors add a section deriving these gradient forms to the appendix please?**
>
> The prefactor $1/2$ arises from two factors: averaging over $G=2$ samples and $H=2$ steps, yielding $1/(GH)=1/4$, and summing over both orderings of each mixed pair (i.e., $(y^{(1)},y^{(2)})$ and $(y^{(2)},y^{(1)})$), yielding a factor of $2$. Together this gives $2/(GH)=1/2$. We added a step-by-step derivation of the GRPO and SGPO gradients to Appendix B.
>
> 9. **(Theory) Would it be possible to add derivations for equations (3) and (4) in the appendix of the paper?**
>
> Yes. We added full derivations of Eqs. (3) and (4) in Appendix B.

---

> ### Author Response · Authors · 2026-01-11
> **Author response to reviewer (part 3/4)**
>
> 10. **(Presentation) Regarding reference solutions: to clarify, must there be a reference solution for every question in the dataset on which the model is trained? I.e. SGPO requires training datasets with fully answered train of thought solutions? We need a reference solution, but not necessarily a full chain of thought, just need to let judge know how to solve this question.**
>
> No. SGPO assumes access to a per-question verification anchor for the judge (e.g., a gold final answer, a brief outline, or a full reasoning trace when available). The anchor enables step-wise error localization without requiring the judge to solve the problem. In our experiments, we use SFT reference solutions with reasoning traces to ensure reliable first-error identification.
>
> 11. **(Presentation) What do "avg@16" and "pass@16" mean? It could be good to briefly describe these.**
>
> We clarified this in Section 4.1. With $k$ samples per prompt, pass@k counts a prompt as solved if any sample is correct, whereas avg@k is the average fraction of correct samples among the $k$ outputs.
>
> 12. **(Presentation) I wonder if the "related works" section could be more logically organized --- it seems like the text jumps back and forth between concepts a little bit. For example, chain of thought reasoning is referenced before the "chain of thought and its variants" section, which makes me wonder if the chain of thought section should come earlier.**
>
> We have reorganized the Related Work section by moving the "Chain-of-thought and its variants" discussion earlier and smoothing the overall flow.
>
> 13. **(Presentation) Perhaps it is just me, but I am somewhat confused by the choice of "colouring" in the title. What is the rationale for this?**
>
> "Coloring" is a metaphor for step-wise annotation of incorrect reasoning: the judge highlights the first error while preserving partial credit for preceding correct steps, which captures the core mechanism of SGPO.
>
>
> ## Minor Issues
>
> 1. **[Intro line 8 and elsewhere]: There are many places in the paper where the authors add "the" where it is not needed. For example, line 8 of introduction: "As the generative AI applications..."**
>
> We have removed redundant definite articles "the".
>
> 2. **[Intro line 9]: "conversational interfaces" is a very strange phrasing; suggest reword.**
>
> We have changed it to "single-turn chat and question-answering".
>
> 3. **[Intro line 9]: "increasingly powerful" $\implies$ "more powerful"**
>
> We have changed it to "more powerful".
>
> 4. **[Intro line 10]: what does "...positioning them as a key frontier in practice" mean? Suggest rephrasing this**
>
> We have changed it to "foundational component of modern AI systems".
>
> 5. **[end of page 1]: first mention of "group-relative" training. I suggest including a quick description of this**
>
> We have added a brief explanation "normalizing rewards across multiple samples for the same prompt" since more formal description is right after this paragraph.
>
> 6. **[page 2, top of 3rd paragraph]: "The key insight is that many reasoning tasks possess a structure where step-level correctness can be explicitly defined." The key insight where? In the above paragraph or in your own work?**
>
> We agree that the phrasing was ambiguous; we have revised "The key insight" to "A common observation" to make clear that we are not claiming novelty for it.
>
> 7. **[first line, page 3]: distinguish $\Rightarrow$ distinguished**
>
> We have changed "distinguish" to "distinguished".
>
> 8. **[section 2]: could be useful to start by mentioning the focus on fine-tuning.**
>
> We have added a paragraph to introduce the standard finetuning process for the readers.
>
> 9. **[3rd equation in section 2]: define "clip" operation**
>
> We have added the definition right after the 3rd equation in section 2.
>
> 10. **[Equation 2]: Why use the sigmoid function? If RTS is zero, shouldn't the reward be zero?**
>
> We clarify the purpose of the sigmoid function in Weakness 3. Recall that we set $\beta=10$ and $\gamma=0.5$, which makes the reward at $\texttt{RTS}=0$ extremely close to zero (specifically, $0.006$). After group normalization, this difference becomes negligible, so the reward is effectively $0$ when $\texttt{RTS}=0$.
>
> 11. **[Example in Remark 2]: the example starting with "The twelve letters A, B, C, D, E, F, G, H, I, J, K, L are randomly grouped" seems redundant.**
>
> We agree that this example is redundant and have removed it from Remark 2.
>
> 12. **[2nd line after first equation section 3.2]: in the definition of the score should be a subscript, right?**
>
> It should be a subscript. We have fixed that.
>
> 13. **In some of the results tables it could be useful to present results as bar plots. The tables are quite dense and difficult to take in at times.**
>
> We add a new visualization in Figure 2, showing the standard deviation and mean results of multiple-run performance for ablation analysis towards judge model setup.

---

> ### Author Response · Authors · 2026-01-11
> **Author response to reviewer (part 4/4)**
>
> 14. **[Table 5]: what does “SGPO/GRPO”, and vice versa, mean in the table heading? Somewhat confusing.**
>
> In Table 5, "SGPO/GRPO" denotes the number of questions solved by SGPO but not GRPO, and "GRPO/SGPO" denotes the number solved by GRPO but not SGPO. We have revised the table hcaption to make this explicit.
>
> 15. **[page 23]: what do you mean by "given that the transition is deterministic"? Are deterministic transitions required for DPO?**
>
> Deterministic transitions are not required by the DPO algorithm itself. In the cited work, it is assumed as part of the token-level MDP formulation of autoregressive LLM generation, which is used in their theoretical development (see, e.g., Lemma 1). We have revised the text to clarify that determinism is not a requirement of DPO.
>
> 16. **[page 23]: when you say "the specific loss using a prospect theory...", what do you mean by "specific loss"? Do you mean the DPO loss?**
>
> Here, "specific loss" refers to the prospect-theory-inspired DPO-style loss variant introduced in the cited paper, rather than the original DPO objective. We have revised the wording to make this explicit and avoid confusion.
>
>
> ## Broader Impact Concerns:
>
> **In addition to potential societal benefits of large AI models, there is a growing body of literature observing certain, seemingly severe, negative impacts—including energy consumption, power/wealth concentration, job loss, and threats to democracy. For easily digestible overviews of such impacts, one could look at the Atlas of AI by Kate Crawford or The AI Con by Hanna and Bender. For academic articles one could start with works by the same authors. Given this growing literature, and given the focus of the article on improving large reasoning models, I was surprised to see no mention of broader impact concerns in the paper. I would like to see at least some brief discussion of broader impacts in the paper before I can recommend it for publication.**
>
> Thank you for raising this important point. We agree that large AI models can entail significant negative externalities. We have added a brief Broader Impacts discussion at the start of Appendix to acknowledge these concerns in the context of improving large reasoning models, and outlining relevant limitations and considerations.

---

> > ### Comment · Reviewer_UtfL · 2026-02-06
> > **reply to authors rebuttal**
> >
> > Thank you very much for the comprehensive responses! I have a few last comments in response to some of your replies, along with notes on some minor typos in the math that I subsequently noticed. Hopefully these may be of use regardless of whether you publish your paper here or at another venue (note that numbering corresponds to your above replies):
> >
> > Replies to responses
> > 1. I understand that you are comparing with GRPO because your work builds most directly from GRPO. However, when a new method (like SGPO) is proposed, to be relevant to the machine learning community it should, in my view, improve upon current best, or standard, practices in some way. To my understanding, GRPO, as a method, has become somewhat out of date—in part because of the issues you bring up here—and has been largely superseded by (or augmented with) process reward models. These process reward models already aim to address the main issue focused on in your paper: to train better multi-step reasoning models by providing partial rewards based on process. Therefore, it seems to me that it would be more relevant to compare with other modern approaches (i.e. process reward models) that are aiming to tackle the same problem that you are aiming to tackle with SGPO, rather than comparing with GRPO.
> >
> > 5. I might suggest using terms other than “filter” and “smoothing” for the $\beta$ and $\gamma$ parameters. Given their use for inference in the time-series literature, using these terms might lead to some confusion.
> >
> > 13. Thank you for adding a bar plot! I think this really helps with data presentation. I have just three extra comments about this: (1) the figure is not referenced anywhere in the main text. For it to be useful it would be good to incorporate it into the flow of the text; (2) I suggest using error bars to show standard deviation instead of a second y-axis. Error bars are more standard and easier to interpret; (3) it would be great to have a bar plot that compares GRPO and SGPO, instead of solely different judges for SGPO.
> >
> > Broader impact concerns: great job on the re-write of this section!
> >
> > Comments on proof
> > 1. Lemma B.1 proof equation 1: I think the first plus sign should be a minus, in the denominator
> > 2. Lemma B.1 proof equations 3 and 4: missing $q$’s
> > 3. Theorem 3.3 proof equation 1: $\Delta_{SGPO}^{(k)}$ is not yet defined in the main text, as far as I can tell
> > 4. Theorem 3.3 proof, second set of equations after table 1: I believe the factors of $1/2$ in the denominator should be just $1$.
> > 5. Theorem 3.3 last line of proof: middle two expressions should be squared.

---

> > > ### Author Response · Authors · 2026-02-10
> > > **Author response to reviewer (follow-up 1/1)**
> > >
> > > Dear Reviewer UtfL,
> > >
> > > We again sincerely appreciate your time for the continued feedback over these helpful points.
> > >
> > > For (1), we appreciate the perspective that, for relevance to the broader ML community, a new method should ideally be compared against current best practices, including modern PRMs. Our choice to center the comparison on GRPO was deliberate for the following reasons, which we will make clearer in the revised manuscript:
> > >
> > > 1. SGPO is designed as a drop-in modification of GRPO-style group-relative RL, targeting a specific and well-documented failure mode: all-negative-sample groups. Our goal is not to replace PRMs, but to improve learning within outcome-based, GRPO-like pipelines that are widely used in practice (e.g., DeepSeek-R1-style training).
> > >
> > > 2. PRMs require either (i) prefix-level value estimation or (ii) dense step-level supervision that approximates a value function. In contrast, SGPO uses post-hoc first-error localization without predicting future success or guiding exploration. This distinction is important both algorithmically and theoretically, and we will emphasize more clearly that SGPO is complementary to PRMs rather than a competing PRM variant.
> > >
> > > 3. Theoretical focus. Our theoretical results analyze GRPO-style group-normalized policy gradients. Extending this analysis to PRMs (which fundamentally alter the objective and credit assignment mechanism) would require a different formal framework and is beyond the scope of this paper.
> > >
> > > That said, we agree that the relationship to PRMs should be made more explicit. In the revision, we will strengthen the discussion contrasting SGPO with PRMs, clearly positioning SGPO as an orthogonal and complementary approach rather than a replacement. We will also include preliminary experiments exploring the interaction between SGPO-style negative-sample differentiation and PRMs, and add a forward-looking discussion on how the two paradigms can be combined in future work.
> > >
> > > For (2), we agree and have rephrased the expression in the first paragraph under section 4.3.
> > >
> > > For (3), regarding the 1st point, we agree and have included a reference to the Figure; regarding the 2nd point, since variances are fluctuated within a very small range, presenting it in error bar format wouldn't make it clearly distinguish betwene each, we therefore choose to design a double-axis for it seperately; regarding the 3rd point, we agree and will take time to incorporate it into the Figure for the final version.
> > >
> > > We have revised the entire proof. Many thanks for your close reading and helpful suggestions. We believe the manuscript has improved significantly thanks to your valuable feedback.
> > >
> > > ---
> > >
> > > Sincerely,
> > >
> > > Authors of Paper 6651

---

### Review · Reviewer_4mMY · 2025-12-21

**Summary Of Contributions:**

## Summary

* GRPO samples a group of trajectories to compute advantage. However often times early on during training, it is possible that none of the policy rollouts in a given group end with the correct state. When this happens in an RLVR training situation no gradient update takes place leading to a wasted step.
* This paper (SGPO) proposes modification to GRPO where a new equation for computing advantages is used which allows for gradient updates even when the entire group consists of negative samples (ie. rollouts that result in the incorrect answer).
* The key mechanism for SGPO is to use a judge model (aided by a ground-truth reference response) which examines the policy rollout if it is negative and assigns a score (called Reasoning Trajectory Score or RTS) to incorrect rollouts. This score is computed based on what percentage of the reasoning trace was correct (by finding the first error and considering all prior actions correct).
  * The RTS can be computed by a separately trained model or by an existing LLM, in this paper they simply use existing LLMs
  * When using an existing LLM the technique is performing a kind of distillation since it is using an existing trained LLM but it doesn't follow the traditional knowledge distillation formulation since the "teacher" model doesn't need to produce correct answers (therefore it's not really a "teacher" and really acts more like a "judge").
* The main strengths of this paper & the method are that it finds a way to take advantage of all negative groups in GRPO which allows for sped up convergence and faster training. The authors provide a simplified setting experiment in Figure 1 which clearly shows the improved learning dynamics from SGPO in theory.

**Additional Comments:**

I'm not as familiar with the literature for RLVR in LLMs and thus am unable to adequately judge the novelty of this contribution in its entirety. Based on a cursory understanding the paper does not incorrectly claim any novelty and the authors make clear the distinctions from prior work in their remarks, however due to my lack of knowledge about all prior work I cannot evaluate this for certain beyond my brief literature review.

**Audience:**

Yes

**Audience Explanation:**

Yes this paper is of clear interest to the TMLR community. It proposes a new mechanism for computing advantage in RLVR training which is able to use groups of rollouts where all members are incorrect. This fits with the broader themes called for by TMLR papers and the findings are properly supported through both theory and experimentation.

**Broader Impact Concerns:**

Given the focus on more efficient training with faster convergence it would be interesting for the authors to compare the extra compute needed to run the judge model vs the compute saved by the faster convergence of SGPO. The authors mention a 10% wall-clock time overhead for the judge model. I would like to see a more explicit trade-off analysis: Does the accelerated convergence of SGPO reduce total training compute by _more_ than this 10% overhead? Currently, it is unclear if SGPO is more compute-efficient net-net. Ideally the small amount of compute needed for a judge model is offset by the improved convergence properties (though I recognize this doesn't take into account the cost of training the judge model).

**Claims And Evidence:**

Yes

**Claims Explanation:**

In general I believe that most of the authors claims are supported. Specifically that SGPO provides faster convergence compared to GRPO seems to be supported well by their various experiments. Additionally it is clear from the final benchmark numbers on various mathematics benchmarks that SGPO does not degrade performance. However in section 4 the authors also claim that SGPO can provide stronger performance than GRPO and I am not 100% convinced on this claim by the evidence. The overall performance of SGPO models seems to be quite similar to those trained with GRPO and based on the main results in Table 2 it seems to provide marginal if any improvements. I believe the authors are focusing on Table 5 to make their claim of "stronger performance" which focuses on pass@16 vs the pass@1 and avg@16 reported in Table 2.

**Requested Changes:**

In order to better support the claim that models trained with SGPO are "stronger" simply reporting pass@16 for a subset of benchmarks is insufficient. Looking at Table 2, the gains for 8B, 14B, and 32B models are all <1% in the 'Overall' metric. The authors should clarify if these gains are statistically significant or if the primary benefit is strictly convergence speed. If the primary benefit is convergence speed then the "stronger" claim should be removed from the manuscript.

---

> ### Author Response · Authors · 2026-01-11
>
> Thank you for the careful reading and concrete suggestions. We agree that the manuscript should more clearly distinguish between benefits in convergence speed and final performance, and we have revised the paper accordingly.
>
> 1. **However in section 4 the authors also claim that SGPO can provide stronger performance than GRPO and I am not 100% convinced on this claim by the evidence. The overall performance of SGPO models seems to be quite similar to those trained with GRPO and based on the main results in Table 2 it seems to provide marginal if any improvements. I believe the authors are focusing on Table 5 to make their claim of "stronger performance" which focuses on pass@16 vs the pass@1 and avg@16 reported in Table 2.**
>
> We agree that our original wording was imprecise. The main empirical takeaway is that SGPO achieves broadly comparable final performance to GRPO on standard metrics (Table 2), with modest gains in some settings rather than a universal improvement. Table 5 was intended as a complementary analysis: pass@16 and unique-solved counts highlight cases where SGPO solves additional hard problems that GRPO does not, even when average differences in Table 2 are small. In the revision, we have replaced the phrase "stronger performance" with more precise language emphasizing comparable final performance and improved coverage on harder problems in some settings.
>
> 2. **In order to better support the claim that models trained with SGPO are "stronger" simply reporting pass@16 for a subset of benchmarks is insufficient. Looking at Table 2, the gains for 8B, 14B, and 32B models are all $<$ 1% in the 'Overall' metric. The authors should clarify if these gains are statistically significant or if the primary benefit is strictly convergence speed. If the primary benefit is convergence speed then the "stronger" claim should be removed from the manuscript**
>
> We agree. The differences below 1% in the "Overall" metric in Table 2 are not always statistically significant, as most results are based on single runs under a fixed compute budget. Accordingly, we have removed the "stronger" wording. The primary practical benefit we emphasize is faster convergence (Figure 3), together with improved pass@16 and unique-solved coverage on harder benchmarks (Table 5). The manuscript has been revised to reflect this more accurately.
>
> 3. **Given the focus on more efficient training with faster convergence it would be interesting for the authors to compare the extra compute needed to run the judge model vs the compute saved by the faster convergence of SGPO. The authors mention a 10% wall-clock time overhead for the judge model. I would like to see a more explicit trade-off analysis: Does the accelerated convergence of SGPO reduce total training compute by more than this 10% overhead? Currently, it is unclear if SGPO is more compute-efficient net-net. Ideally the small amount of compute needed for a judge model is offset by the improved convergence properties (though I recognize this doesn't take into account the cost of training the judge model).**
>
> This is an excellent suggestion. While enabling SGPO introduces approximately a 10% wall-clock overhead due to judge model calls, in practice, SGPO is used only as an early-stage accelerator: we apply step-wise supervision only during the first three epochs, after which training reverts to standard GRPO. Since the largest convergence gains occur early (Figures 3 and 4), the reduction in total training iterations needed to reach a target performance typically offsets the additional per-step overhead. As a result, the net compute cost is comparable or lower in our setting. We have added a clarification of this trade-off to the revised manuscript.

---

### Review · Reviewer_yPcN · 2025-12-27

**Summary Of Contributions:**

This paper proposed a more fine-grained reward system for reasoning RL training that has a higher probability to distinguish responses for harder prompts to avoid "all-negative and equal" scores. It conducts empirical experiments together with theoretical analysis to demonstrate the effectiveness of the proposed method. Here are some key pros and cons:

Pros
* The method is relatively easy-to-implement and seems effective. In general, the method is quite intuitive and results are reasonable. The experimental results shown in this paper can be a reference for future studies.

Cons
* The proposed method is an intuitive yet straightforward extension to existing methods. In fact, it is only applicable in certain conditions that I am hesitated to call it a systematic method. First, it is pretty intuitive to see that when "all negative" signals occur there are at least 2 options you can do: (1) improve your reward system so that it is doing a better job distinguishing bad examples, or (2) design a whole new algorithm system that works differently from the existing RL. The proposed method falls into the first category, which is very reasonable but the results then become expected, as long as the reward is reliable. So I feel the baseline here is not really GRPO but rather the reward system used in solving Math/STEM problems. Yet the design of reward system is a creative and complicated topic, and I feel the proposed method is not abstract enough to generalize to all reward methods. Second, there are many limitations to the method proposed. For example, it currently only improves on step-wise reasoning models that are trained on rewards computed purely on final answers. It also relies on stronger models for reference and correct judgements for reasoning-step correctness. And most importantly, it has the same issue of the original system if all samples fails at the same steps, which results in same scores again.

Overall, I believe that the method is effective and I think it is great for other researchers to see this results. However, I feel the method is only proposed for and studied under a set of restrictions, so it might be better to carefully position the true contribution of this paper.

**Audience:**

Yes

**Audience Explanation:**

See above

**Claims And Evidence:**

Yes

**Claims Explanation:**

The paper contains plenty of experimental and theoretical results.

**Requested Changes:**

None

---

> ### Author Response · Authors · 2026-01-11
>
> Thank you for your careful reading and constructive feedback. We appreciate the opportunity to clarify the scope, baselines, and limitations of our approach. Below we respond point by point and indicate corresponding clarifications made in the revised manuscript.
>
> 1. **The proposed method is only applicable in certain conditions; it is hard to call it a systematic method, and it may not generalize to all reward methods.**
>
> We agree with this assessment and do not intend to position SGPO as a universally applicable reward-learning framework. SGPO is designed for a specific yet practically important regime: structured step-wise reasoning tasks trained with outcome-based (final-answer) rewards, where all-negative-sample groups are common and step-wise error localization can provide meaningful partial credit. To avoid any overgeneralization, we have added an explicit scope statement at the end of the paragraph immediately preceding Contribution, clarifying both the intended setting and the limitations of applicability.
>
> 2. **The baseline may not be "GRPO" but rather the reward system for Math/STEM; the proposed method appears to be a straightforward reward-design extension.**
>
> We agree that SGPO introduces a new reward construction. Our choice of GRPO as the primary baseline is deliberate: SGPO is designed as a drop-in modification to the GRPO pipeline. Specifically, we keep the rollout procedure, policy update rule, and outcome supervision unchanged, and only replace the binary outcome reward with a step-wise–shaped scalar reward when computing within-group advantages. This controlled comparison isolates the effect of reward shaping within the GRPO framework, which is the intended deployment setting of SGPO. We have added a clarifying sentence to this effect in the paragraph immediately above Remark 3.1.
>
> We also agree that other Math/STEM reward systems are relevant baselines. A systematic comparison with alternative reward designs (e.g., process-reward models) would be valuable, but requires substantial additional experimentation and is beyond the scope of this manuscript. We view this as an important direction for future work.
>
> 3. **The method currently only improves step-wise reasoning models trained with rewards computed purely on final answers, and may not be abstract enough to generalize.**
>
> We agree. Our current study intentionally focuses on outcome-based training regimes for structured step-wise reasoning tasks, where final-answer rewards are sparse and all-negative-sample groups are prevalent. In our setting, step-wise error localization can transform otherwise uninformative negative samples into graded learning signals. We agree that SGPO can not be directly generalized to arbitrary reward settings. We have added an explicit scope statement in the revised manuscript to make this limitation clear, and we identify extensions beyond outcome-based rewards as future work.
>
> 4. **The proposed method relies on stronger models for reference and correct judgements for reasoning-step correctness.**
>
> This is an important point. SGPO does not require the judge model to solve the problem from scratch, nor does it require the judge to be strictly stronger than the base model. The judge is provided with a reference anchor (e.g., a solution outline or reasoning trace) and a candidate rollout, and its role is limited to identifying the first clear inconsistency relative to the reference. Empirically, we observe that neither the base model nor the judge model can reliably solve certain hard problems on their own; nevertheless, the combination of the base model’s attempts and SGPO’s step-wise error localization yields informative training signals that enable learning to progress (see Table 5). We have clarified this distinction in the revised manuscript.
>
> 5. **The method may still suffer from the same issue as GRPO if all samples fail at the same step, resulting in identical scores again.**
>
> We agree that this is a remaining limitation. If all trajectories in a group fail at the same first-error position, they receive identical $r_{\texttt{SGPO}}$, and group-normalized advantages can still vanish, as in GRPO. Our claim is therefore that SGPO alleviates (but does not fully eliminate) this degeneracy by making such exact ties significantly less likely in practice. We have explicitly added this limitation to the end of Remark 3.1.

---

### Author Response · Authors · 2026-01-11
**Author response submitted & updated manuscript**

We thank all the reviewers for your time and for the constructive, insightful feedback. We have responded to all questions and updated the manuscript; non-trivial changes are highlighted in green.

---

Sincerely,

Authors of Submission 6651

---

### Comment · Action_Editor_9qTK · 2026-02-20
**Question for the authors**

The current abstract contains the following text:

> We also empirically validate the proposed stepwise guided policy optimization (SGPO) method, demonstrating consistent gains across model sizes (7B, 14B, 32B) in offline and online training on 9 benchmarks, including base and distilled variants

Do you still believe this, despite reviewers UtfL's and 4mMY's comments?

Specifically the second point of Weakness 2 in UtfL's review:

> It does not seem that the authors’ model, SGPO, universally outperforms GRPO—or even the baseline—in terms of performance (see e.g. Table 2). One would have expected, at first glance, that having the extra information from all-negative samples should, at worst, not drop below the baselines that do not address the negative sample issue. However, there is not much discussion of this in the paper. It would be good to have some deeper discussion of when and why the authors’ approach does not outperform these alternatives.

Seems to not have been addressed in any response, and the text has not been changed to acknowledge this.

Please comment on this directly in a reply to this post.

---

> ### Author Response · Authors · 2026-02-20
> **Clarifying consistent gains**
>
> Thank you.
>
> We agree with reviewers UtfL and 4mMY that the original abstract wording "consistent gains" was too strong and can be misread as claiming uniform dominance over GRPO/baselines. We no longer make that claim in the revised paper. We changed the abstract sentence to explicitly reflect what Table 2 shows: *Overall, SGPO improves average performance and is effective in early and mid-training when all-negative groups are prevalent, while improvements are not uniform across every benchmark and depend on the structure and informativeness of negative samples*.
>
> We also added a discussion near the training/online-results section (around the Table 2 discussion) to address when and why SGPO may not outperform GRPO/baselines: *That being said, these benefits are not uniform across every benchmark. When negative samples are short, highly noisy, or dominated by early derailments, step-wise judging provides less actionable signal and can even introduce additional variance through judge noise, so SGPO may match or occasionally underperform GRPO or the baseline on specific tasks. This pattern is consistent with our empirical results and motivates a more nuanced view: SGPO is most effective when the model’s failures retain informative intermediate structure*.
>
> In Section 4.4, we have added a discussion: *We emphasize that SGPO does not guarantee uniform improvements on every benchmark: its gains depend on the structure of negative samples. SGPO is most beneficial when failures are due to truncated but largely correct trajectories or localized mistakes, whereas when negative samples fail very early or lack meaningful structure, the step-wise signal can be less informative and gains may diminish*.
>
> To answer your question directly: no. We do not still believe the "consistent gains" wording, and we have revised both the abstract and the paper’s discussion to acknowledge the non-uniform per-benchmark behavior and explain the mechanisms (negative-sample structure + judge noise/variance) behind it.

---

### Decision · Action_Editor_9qTK · 2026-02-20

**Recommendation:** Accept with minor revision

**Additional Comments:**

Two of three reviewers recommended to accept, and I believe the main point about claims & evidence of the reject recommendation has been resolved in the most recent paper update (post-recommendations).

However, this means that there may be other such issues that remain unaddressed. The authors have shown that they are willing to modify the wording in regards to comments made about claims & evidence by reviewers. Hence, I recommend acceptance with minor revision.

The revision should address any outstanding issues related to claims & evidence that were brought up in reviews. I will ask the reviewers to check over the final draft and verify any outstanding to ensure that they are adequately addressed.

**Audience:**

Yes

**Audience Explanation:**

Second reason for reject recommendation:
>  [...] because of the benchmarking issues—issues also noted by the other two reviewers. For a new method to be relevant for the community I believe that proper benchmarking needs to occur, and I do not think that the study has achieved this.

Again, there was a question of whether this will be interesting "enough" to the community given that the results are not consistent gains across the board and that there proper benchmarking is required.

However, proper benchmarking is subjective. E.g. a comparison to PRMs would indeed make the results stronger. However, since GRPO is only a year old, I don't think it is necessary to compare to newer techniques in order to satisfy the TMLR acceptance criterion for interest, as long as the evidence supports the claims.

The topic itself is of interest to TMLR and a comparison of GRPO is sufficient for interest to TMLR audience.

**Claims And Evidence:**

Yes

**Claims Explanation:**

There was some remaining uncertainty from one reviewer about the claims as presented in the paper post-rebuttal, but this has now been resolved by the most recent edit based on the question I asked the authors.

From one critical recommendation, the reviewer recommends rejection based on two main reasons:

> I have voted to reject on account of the claims not being support by the results, as discussed above. I also have some other issues with the paper around presentation, and around the relevance of the studied theoretical framework—specifically, that it is to simplified to provide deep insight—but my primary concern is around benchmarking.

The particular claim not supported by the evidence has now been resolved.

(Second reason below.)

The other two reviewers were both positive, and one in particularly stated that the claims are supported by the evidence.

---

> ### Author Response · Authors · 2026-02-25
>
> Dear AE,
>
> Thank you very much for your time and support. We fully agree with the wording suggestion and have carefully revised the claims regarding contributions and supporting evidence. All changes have been incorporated into the camera-ready version we just uploaded. Specifically:
>
> * **Page 3 (second contribution):** We added a sentence clarifying that our scope comparison baseline targets outcome-based verification, rather than implicit PRMs.
> * **Page 10 (bottom paragraph):** We added an explanation to strengthen the evidence supporting the contribution of negative samples.
> * **Conclusion:** We added a clarification acknowledging the issue of "uniform gain": *"Empirically, SGPO improves average performance and improves coverage on harder problems, with the largest benefits in early and mid-training when all-negative groups are prevalent; gains are not uniform across benchmarks and depend on the structure/informativeness of negative samples."*
> * **Section 4.4:** We added two additional paragraphs to clarify the benchmark scope.
>
> We believe these revisions make our claims and supporting evidence clearer. We again thank the reviewers and you for your time and helpful suggestions.
>
> ---
>
> Sincerely,
>
> Authors of Paper 6651

---

> > ### Comment · Action_Editor_9qTK · 2026-03-04
> > **Thank you**
> >
> > I appreciate the extra edits. I am going to ask one reviewer to take one last look. Please allow a few days for this, thanks.

---

> > ### Comment · Action_Editor_9qTK · 2026-03-10
> > **Possible typo?**
> >
> > In discussion about the camera-ready, one reviewer pointed out:
> >
> > > Note that I found what seems to be a minor typo on page 10: when the authors define what “pass@k” means they say "Here, pass@k is the percentage of prompts solved by at least one sample”. I think they mean "Here,
> > pass@k is the percentage of prompts solved by at least k samples”.
> >
> > Please confirm.

---

> > > ### Author Response · Authors · 2026-03-10
> > >
> > > Hi AE,
> > >
> > > Many thanks for pointing it out. We have corrected to "Here, pass@k is the percentage of prompts for which at least one of the k sampled responses is correct" in the camera-ready version just uploaded.
> > >
> > > ---
> > >
> > > Sincerely,
> > >
> > > Authors of Paper 6651